# Towards Understanding Text Hallucination of Diffusion Models via Local Generation Bias

**Rui Lu**[1*], **Runzhe Wang**[2*], **Kaifeng Lyu**[3], **Xitai Jiang**[4], **Gao Huang**[1], **Mengdi Wang**[2]
[1]Department of Automation, Tsinghua University
[2]Electrical and Computer Engineering, Princeton University
[3]Simons Institute, UC Berkeley
[4]Qiuzhen College, Tsinghua University
{r-lu21, jiang-xt21}@mails.tsinghua.edu.cn
{runzhew, mengdiw}@princeton.edu
kaifenglyu@berkeley.edu, gaohuang@tsinghua.edu.cn

## Abstract

Score-based diffusion models have achieved incredible performance in generating realistic images, audio, and video data. While these models produce high-quality samples with impressive details, they often introduce unrealistic artifacts, such as distorted fingers or hallucinated texts with no meaning. This paper focuses on textual hallucinations, where diffusion models correctly generate individual symbols but assemble them in a nonsensical manner. Through experimental probing, we consistently observe that such phenomenon is attributed it to the network's local generation bias. Denoising networks tend to produce outputs that rely heavily on highly correlated local regions, particularly when different dimensions of the data distribution are nearly pairwise independent. This behavior leads to a generation process that decomposes the global distribution into separate, independent distributions for each symbol, ultimately failing to capture the global structure, including underlying grammar. Intriguingly, this bias persists across various denoising network architectures including MLP and transformers which have the structure to model global dependency. These findings also provide insights into understanding other types of hallucinations, extending beyond text, as a result of implicit biases in the denoising models. Additionally, we theoretically analyze the training dynamics for a specific case involving a two-layer MLP learning parity points on a hypercube, offering an explanation of its underlying mechanism.

## 1 Introduction

Inspired by the diffusion process in physics (Sohl-Dickstein et al., 2015), diffusion models learn to generate samples from a specific data distribution by fitting its score function, gradually transforming pure Gaussian noise into desired samples. These models (Song et al., 2020a; Song & Ermon, 2019; Song et al., 2021; Ho et al., 2020) demonstrate remarkable capability in generating high-quality samples with significant diversity, establishing them as the *de facto* standard generative models for various tasks, including image generation, video generation (Brooks et al., 2024), inpainting (Lugmayr et al., 2022), super-resolution (Gao et al., 2023), and more. However, despite the impressively realistic details produced, diffusion models consistently exhibit artifacts in their outputs. One common issue is the generation of plausible low-level features or local details while failing to accurately model complex 3D objects or the underlying semantics (Borji, 2023; Liu et al., 2023). This phenomenon, known as hallucination, occurs when the generated samples either do not exist in real-world distributions or contain content that lacks semantic meaning. In practice, even large generative models like StableDiffusion (Rombach et al., 2022), trained on enormous datasets, still suffer from these issues—often generating hands with extra, missing, or distorted fingers. In this work, we primarily focus on a special type of artifacts called text hallucinations, where generative model can correctly generate individual symbol in syllabus but assemble them in nonsensical manner. This naturally raises the following question.

---

*Equal contribution, work done during Rui's visit at Princeton

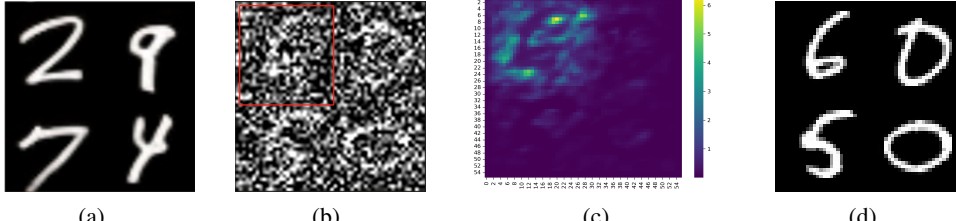

| (a) | (b) | (c) | (d) |

Figure 1: An illustration for local generation bias. We construct a synthetic dataset (a) that all images satisfy the rule that sum of first row equals second row, i.e. 2+9=7+4. Diffusion model starts from noise $x_t$ (b) and using denoising network to generate digit images in four quarters. We found that the top-left region's denoising primarily depends on its own data, depicted by saliency map (c). This means the diffusion model independently generates each digit without caring any other digits, ends up with $x_0$ (d) failing to capture the relation between four digits.

*Why do diffusion models typically struggle with generating images that include text content? How do they learn these distributions and end up generating hallucinated samples?*

This study identifies a critical limitation in diffusion models termed **Local Generation Bias**, where score networks trained via score matching exhibit a tendency to generate outputs based predominantly on localized input regions. This bias leads to uncorrelated symbol generation, as denoising processes for individual tokens operate independently, disregarding inter-token dependencies and underlying rules. To measure the degree of local operation, we propose a probe called the **Local Dependency Ratio (LDR)**, which quantifies the gradient magnitude within the same local region compared with the entire input. A higher LDR indicates a stronger local generation bias. Interestingly, we discover that a high LDR emerges early in training and persists throughout extensive training steps. LDR thus becomes a good indicator for the strength of local generation bias. Moreover, we find this bias is intrinsic in training rather than architectural limitation. Even for models like transformers (Peebles & Xie, 2023; Vaswani, 2017) or MLPs that are designed to capture global dependencies, the local generation bias persists.

To gain a deeper understanding of this phenomenon, we probe into a simple case, providing insights into its underlying mechanism. Specifically, we analyze a two-layer ReLU network learning a distribution supported on the vertices of a hypercube $\{\pm 1\}^d$. This distribution can be among the vertices that satisfy a parity constraint, where the product of all $x$ entries is 1. When fitting the target denoising function for this distribution, we find that the network has certain training bias, inducing it to separately learn $d$ univariate target function for marginal distributions on each dimension, sampling independently over $\{\pm 1\}$ for each entry. Eventually, the generation process samples uniformly over the entire hypercube rather than parity subset, where hallucination happens. This introduces an instance for how training bias may result in hallucinatory generations, and offers insights into hallucinations across other domains and modalities.

In summary, this paper contributes in three key folds.

**Identification of Local Generation Bias**: We define and analyze the phenomenon of local generation bias in diffusion models, which leads to artifacts like text hallucinations.

**Mechanistic Explanation:** We provide a detailed theoretical and empirical investigation into the causes of hallucinations, revealing that they stem from the implicit bias in the training process.

**New Analytical Tools:** We introduce the Local Dependency Ratio (LDR) as a measure of local bias and apply it to explore the diffusion model's behavior across training stages.

## 2 RELATED WORK

**Diffusion Model.** Diffusion models, initially introduced by Sohl-Dickstein et al. (2015), are probabilistic generative models that iteratively add and remove noise from data. Early work Ho et al. (2020) laid the foundation and proposed Denoising Diffusion Probabilistic Models (DDPM) Ho et al. (2020), which significantly improved sample quality and stability. Song et al. (2020b) also

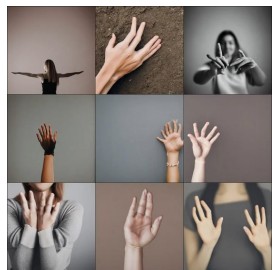 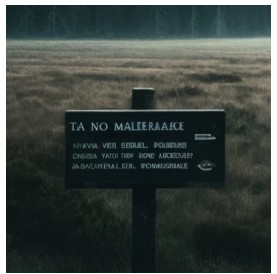 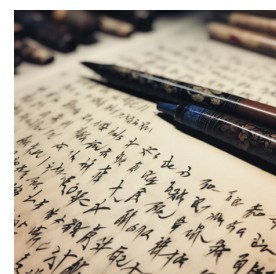

Figure 2: Some examples of deformed hands artifacts and text hallucination in images generated by StableDiffusion Rombach et al. (2022) and Midjourney. Images from prompting "woman showing her hands", "a road sign in a grassland" and "a Chinese traditional calligraphy art".

proposed Score-Based Generative Models (SGMs), unifying diffusion models with other generative frameworks. To address efficiency, Song et al. (2021) introduced Denoising Diffusion Implicit Models (DDIM), reducing sampling steps without quality loss. Diffusion models have since been applied beyond image generation, including video generation Brooks et al. (2024), text-to-image models Rombach et al. (2022), and audio synthesis Kong et al. (2020). Despite advancements, challenges remain, particularly in improving sampling speed and generalization to unseen data, as highlighted by recent experiments in video generation and physics-informed modeling.

**Hallucination in Language Generative Models.** Hallucinations in large language models (LLMs) are a significant challenge, particularly in safety-critical applications, where factually incorrect or logically inconsistent outputs can have severe consequences Ye et al. (2023); Zhang et al. (2023). LLMs may generate erroneous facts, misinterpret instructions, or introduce entirely new information not present in the input, a phenomenon known as input-conflicting hallucination Zhang et al. (2023). Mitigating these hallucinations has become a focus of research, with strategies such as enhancing models with factual data Gunasekar et al. (2023) and integrating retrieval-based mechanisms to ground responses in external knowledge Ram et al. (2023).

**Hallucination in Diffusion Models.** One common artifact of diffusion models is the generation of distorted or deformed body parts, such as hands and legs, which is frequently observed in models like Stable Diffusion Rombach et al. (2022) and Sora Brooks et al. (2024). Additionally, diffusion models struggle with learning rare concepts, particularly those with fewer than 10,000 samples in the training set Samuel et al. (2024). Other common failure modes include models neglecting spatial relationships or confusing attributes, as discussed in prior research Borji (2023); Liu et al. (2023). These issues highlight the limitations of diffusion models when tasked with generating realistic, complex scenes, especially when dealing with rare data or intricate spatial compositions. Recent work Aithal et al. (2024) explains the hallucination of diffusion model via the perspective of mode interpolation, arguing that the improper interpolation between modes yields non-zero density between them, which is the main cause for hallucination.

## 3 NOTATIONS

Denote set $\{0, 1, 2, \ldots, n-1\}$ as $[n]$. To compute the cardinality of a set $S$ we write $|S|$. For a vector $\boldsymbol{x}$, we use $\boldsymbol{x}^{(i)} = \boldsymbol{x}^\top \boldsymbol{e}_i$ to denote its $i_{th}$ dimension, and we use $\boldsymbol{e}_i$ to denote the unit vector along the $i$-th dimension. $\mathcal{N}(\boldsymbol{\mu}, \boldsymbol{\Sigma})$ means a Gaussian distribution with mean $\boldsymbol{\mu}$ and covariance $\boldsymbol{\Sigma}$, $\mathcal{N}(\boldsymbol{x}; \boldsymbol{\mu}, \boldsymbol{\Sigma})$ denotes its density at position $\boldsymbol{x}$. Sampling $x$ from distribution $\mathcal{D}$ is denoted as $x \sim \mathcal{D}$. Asymptotic notation follows the common practice where $f = O(g)$ means there exists a constant $C > 0$ and $x_0$ such that $f(x) < C \cdot g(x)$ for any $x > x_0$. Similarly we write $f = \Omega(g)$ when $f(x) > C \cdot g(x)$ for any $x > x_0$ and $f(x) = \Theta(g(x))$ if $f = \Omega(g)$ and $f = O(g)$. And $*$ stands for convolution operation between two distributions, $f * g(t) = \int_\Omega f(\tau) g(t - \tau) \mathrm{d}\tau$. We use $\Delta(S)$ to denote the set of valid probability distributions over a compact set $S$. We use $\mathrm{sgn}(x) = 1[x > 0] - 1[x < 0]$ to denote the sign function.

## 4 EXPERIMENT STUDY

In this section, we introduce the experimental setup and results of our study on text hallucination in diffusion models. We first reproduce text hallucination phenomenon across different modalities and text rules in our simple synthetic setting. To understand how it originates, a key probe called **Local Dependency Ratio (LDR)** is introduced to quantitatively measure the denoising function's input dependency on local regions. With LDR as a probing tool, we discover the following important observations.

- **High LDR value is always observed when hallucination happens.** This indicates that the denoising model predicts noise by each symbol's region itself, therefore conducting denoising and generation iteration respectively with almost zero entanglement between different symbols. Since the starting Gaussian distribution is also isotropic, the entire generation process for different symbols becomes independent, resulting in incorrect assembly and hallucination.

- **Such phenomenon is ubiquitous across different distributions and architectures**, even for those models with global receptive field such as MLP and DiT(Peebles & Xie, 2023). This indicates such bias is related to ubiquitous implicit bias in training dynamics rather than architectural limitation.

- **As training progresses, LDR decreases and the denoising model starts to overfit**. After extensive training, denoising network overfit to training dataset. This requires it to coordinate different symbols to exact replicate training data, resulting in a drop in LDR.

### 4.1 FORMULATION OF TEXT DISTRIBUTION

The process of constructing a synthetic text-like distribution is as follows. First, we define a set of discrete symbols, $\mathcal{S} = s_1, s_2, \ldots, s_K$, as the syllabus. Next, we define a symbol index list, $\mathcal{I} = (i_1, i_2, \ldots, i_L) \subseteq [K]^L$, which represents a sequence of symbols $(s_{i_1}, s_{i_2}, \ldots, s_{i_L})$. A grammar rule is a probability distribution $P_G$ that defines the validity of a symbol sequence by its probability density, $P_G(\mathcal{I})$. The symbol sequence is then mapped to an ambient space by a function $h : \mathcal{S} \mapsto \mathbb{R}^d$, which assigns each symbol to a vector, such as an image pixel or a scalar. The full signal is formed by concatenating these vectors. Examples of such signals include images with text, time series, and text sequences. We aim to learn the distribution of these signals for generation purposes. For simplicity, we use $h(\mathcal{I}) : \mathcal{S}^L \mapsto \mathbb{R}^{d \times L}$ to represent the rendering process, where a list of tokens is transformed into the input space. We can first sample a token list $\mathcal{I} \sim P_G$ and then apply the rendering function $h$. Throughout the experiments, we fix $L$ to maintain a consistent symbolic system.

In this paper, we mainly test two synthetic symbol assembling rules, including *(i) Parity Parenthesis*, each sample image contains $L$ parenthesis where left symbol "(" and right ")" both have even numbers; *(ii) Quarter MNIST*, each sample image consists of four MNIST digits in the corners and the sum of first row equals the second. More details are in appendix.

### 4.2 TEXT HALLUCINATION RESULTS

After constructing synthetic text distributions $h(P_G)$. The denoising model is trained to fit the score function of these distributions in the ambient space. For embedding vector ambient space, we employ MLP to learn the score function. For image sample, we use modern denoising network including UNet Ronneberger et al. (2015) and DiT (Peebles & Xie, 2023). Note that both MLP and DiT have global receptive field in its function, enabling them to model long range correlations. UNet also embraces attention module in its pipeline.

**Parity Parenthesis.** Our initial attempt starts with parity rule with parenthesis symbol. We fix $L = 8, 16$ and use image of left and right parenthesis to represent symbol $1$ and $-1$. A UNet model (with attention) is trained on this image distribution and learn to generate samples. The details of model architecture is in the appendix. We are interested in whether diffusion model can find clues about parity rule and faithfully reproduce it. An OCR function is utilized to transform the generated image into binary sequences and test whether it satisfies the parity rule. For $L = 8$ we use half fraction of the valid parity images and $5\%$ for $L = 16$. The generated images are categorized into four types, including *(i) Invalid, the low level detail for each symbol is ambiguous and fuzzy hence OCR fails; (ii) Hallucination, each symbol is clear but the overall combination does not fit in rules;*

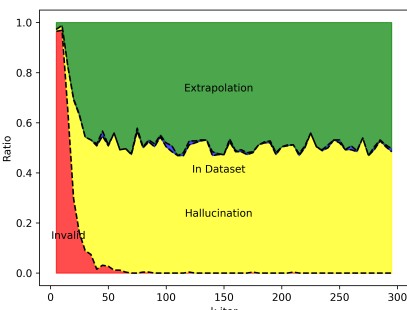 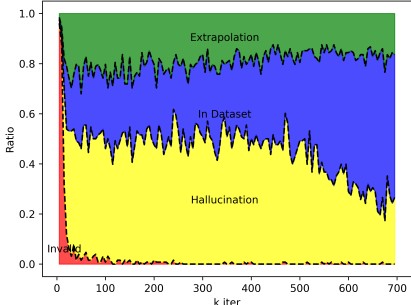

Figure 3: Experimental Result for UNet learning parity parenthesis $L = 16$ (left) and $L = 8$ (right).

*(iii) **In Dataset,** the model exactly reproduce dataset images; (iv) **Extrapolation,** the model generates data sample that satisfy the rule while not presented in the dataset.*

The diagram for different categories' proportion is in figure 3. Note that random guess has $50\%$ chance of satisfying parity requirement. We can see the model quickly learn to generate individual symbol's appearance, and the proportion of *invalid* drops immediately. However, the diffusion model fails to capture the parity rule, half of whose generated images are hallucination. The situation diverges according to the sequence length. In $L = 8$ case the model eventually successfully overfits to the training dataset, but still generates $25\%$ hallucinated samples. For $L = 16$, The model continues to generate correct samples only by chance till the end of training. This simple experiment demonstrates the difficulty for pure-vision based model to learn underlying rule unconditionally. Detailed generated samples is left in appendix.

**Quarter-MNIST:** We also test another symbol system, where four MNIST digit images are assigned in four quarters of an image and satisfy simple arithmetic relations. To achieve low divergence between generation distribution and real distribution, the diffusion model not only needs to generate reasonable digits, but also understands the global relations between these digits.

Simple combinatorics tells there are total 670 combination of symbols $(s_1, s_2, s_3, s_4) \in \mathcal{S}^4$ satisfying $s_1 + s_2 = s_3 + s_4$. We randomly leave out 200 combinations as test set and render the images of the rest. Both UNet and DiT undergo a phase that most of its generated samples do not satisfy the addition requirement, which means hallucination. As the training progresses, both models gradually learn to reproduce sample within dataset. DiT performs better accuracy ($\sim 90\%$) in generating samples satisfying addition relations compared with UNet ($20.6\%$). However, *none of them* is able to generate valid symbol tuple beyond the training dataset, with a fraction only less than $0.5\%$. Therefore there is no extrapolation region in figure 4. In other words, for such text distribution, diffusion model can only struggle between hallucination and overfitting, if no prior knowledge is provided.

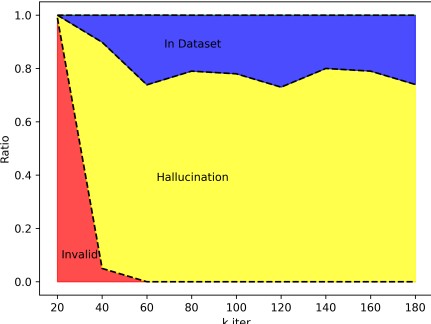 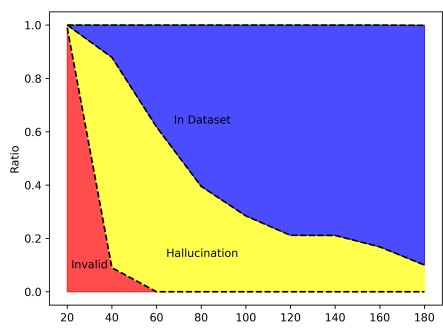

Figure 4: Experimental Result for learning Quarter-MNIST using UNet (left) and DiT (right).

### 4.3 LOCAL DEPENDENCY RATIO ANALYSIS

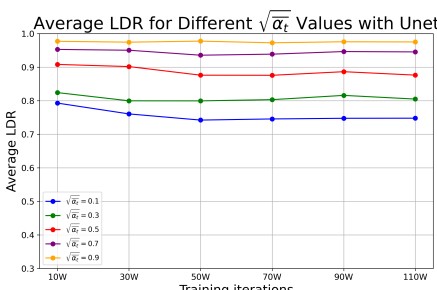 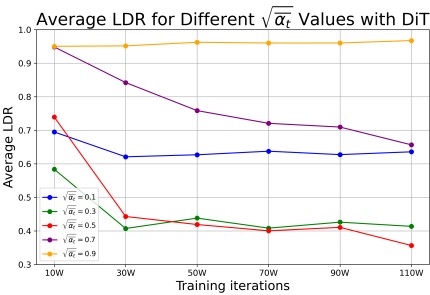

Figure 5: LDR trend for UNet (left) and DiT (right) at different denoising timestep and training iterations. The LDR for UNet remains high throughout the training, therefore it stucks with hallucination. While DiT successfully progress to reduce the LDR, meaning it starts to overfit and memorize the dataset. We select timestep $t$ corresponding to $\sqrt{\bar{\alpha}_t} \approx 0.1, 0.3, 0.5, 0.7, 0.9$.

To investigate the mechanism behind text hallucination. We propose a novel probe called **Local Dependency Ratio**, or LDR in abbreviation. LDR quantitatively measures the degree of the diffusion network that performs denoising and generation locally. Given a trained network $s_\theta(\cdot)$ and certain fixed timestep $t$ with corresponding parameter $\bar{\alpha}_t$. In the total input space $\mathbb{R}^{d \times L}$, denote the region of interest as $\mathcal{R} \subseteq [d \times L]$ referring to the set of entries corresponding to one (or few) symbol's area. Define indicator matrix $P_\mathcal{R} = [e_i]_{i \in \mathcal{R}}$ that filters out entries of $\mathcal{R}$. We compute gradient of the filtered entries $f_{\mathcal{R},\theta}(x) := P_\mathcal{R}^\top s_\theta(x)$ and get Jacobian matrix

$$J_{\mathcal{R},\theta}(x) := \frac{\partial f_{\mathcal{R},\theta}}{\partial x} \in \mathbb{R}^{|\mathcal{R}| \times d}. \tag{1}$$

Define local Dependency Ratio (LDR) function for model $s_\theta$ and region $\mathcal{R}$ as

$$LDR(\theta, \mathcal{R}) = \mathbb{E}_{x \sim p_t} \left[ \frac{\mathrm{Tr}(P_\mathcal{R}^\top J_{\mathcal{R},\theta}(x)^\top J_{\mathcal{R},\theta}(x) P_\mathcal{R})}{\mathrm{Tr}(J_{\mathcal{R},\theta}(x)^\top J_{\mathcal{R},\theta}(x))} \right]. \tag{2}$$

Intuitively, matrix $J$ measures dependency of each output's entry in $\mathcal{R}$ with respect to the input, which is commonly known as *saliency map* (Simonyan, 2013). The difference to conventional saliency map is that each input dimension $x^{(i)}$ receives a gradient vector $g_i \in \mathbb{R}^{|\mathcal{R}|}$ rather a single scalar. It records the sensitivity of output region $\mathcal{R}$ with respect to a certain input entry $x^{(i)}$.

Therefore, $\mathrm{Tr}(J^\top J)$ computes the Frobenius norm of $J$, which is the total sum of all gradient vectors' squared norm. Meanwhile, $JP_\mathcal{R}$ filters dependency gradient within $\mathcal{R}$ itself, thus $\mathrm{Tr}(P_\mathcal{R}^\top J^\top J P_\mathcal{R})$ measures the total summation of squared gradient norms within the same local region $\mathcal{R}$. The LDR is thus within range $[0, 1]$, where a higher value indicates a more local denoising and generation manner.

With LDR, we can probe the model trained on different datasets at various checkpoints. Here we mainly present our probing result for Quarter-MNIST dataset. More visualization and other experimental detail is left to appendix. We select $\mathcal{R}$ to be the top left region, namely the first digit's position, and compute LDR for this region at different denoising steps and training iterations. As shown in figure 5, UNet's LDR remains more than $0.75$ throughout the entire training process, which means it highly focuses on region $\mathcal{R}$ itself to conduct denoising and generation. This could explain why UNet ends up with a much lower accuracy. DiT also presents similar trend, showing a high LDR value at initial stage of training, therefore generating hallucinated samples. However, due to strong approximation power of transformer architecture, its LDR decreases at 30k to 50k iteration, and this synchronizes with the rapid increase of the generated sample's accuracy (see figure 4).

This result provides evidence for the local generation bias. Despite the capacity to modeling global long-range relations, both attention version UNet and DiT appears to rely on information confined within local regions. This is reasonable because in these symbolic systems $P_G$, any two symbol's distribution is independent, namely $P_G(s_i = a, s_j = b) = P_G(s_i = a)P_G(s_j = b)$. Such independence leads denoising network to treat symbols as uncorrelated. As a consequence of such local generation preference, the denoising network separately learns and samples from each symbol token's marginal distribution at early training stage, resulting in text hallucination.

This finding is consistent and universal across different denoising architectures and grammar rules. More experiment details for different distributions are left in appendix. We also visualize $J$ as heatmap of each pixel's gradient magnitude and verify its concentration near the selected region $\mathcal{R}$. For real-world distributions that do not satisfy independent condition, please refer to discussion section.

## 5 DISTRIBUTIONS ON A HYPERCUBE: A THEORETICAL CASE STUDY

In this section, we give an instance that illustrates why high LDR values predict hallucinatory generations, and why the learned score network is biased towards high LDR scores. We consider the case of $|\mathcal{S}| = 2$, e.g. the generation of distributions on the hypercube $\{\pm 1\}^d$, or equivalently on a sequence of $d$ binary tokens. We find that in early training stage neural networks prefer to learn the tokens in their marginal distributions and fails to learn correlations that mark the semantic rules. The theoretical analysis explains why high LDR can be a signal for bad model and defective training process. While the set of score functions with LDR value 1 have restricted generation capacity, under certain grammar rules they form an invariant set of the optimization process. Namely, the gradient component that lower the LDR value vanishes when LDR approaches 1. Therefore in the view of optimization the network will struggle in escaping a high LDR region. Moreover we show that in the instance of a two-layer ReLU network, early training dynamic actually biases the network to a LDR level close to 1 with small initialization, exhibiting the presence of such a struggle in the training process. As a result, the network creates hallucinatory generations.

### 5.1 PRELIMINARIES

#### 5.1.1 PROBABILITY ON THE HYPERCUBE

A spelling/grammar rule $P_G$ over binary sequences specifies a probability distribution function $p_0$ on the hypercube $\{\pm 1\}^d$. We can express $p_0$ in terms of the Fourier expansion on the hypercube, as the indicator functions $x_I = \prod_{i \in I} x_i$ for all possible $I \subset [d]$ forms an orthonormal basis over the uniform distribution $\mathcal{D}$ over the hypercube (namely, $\mathbb{E}_{x \sim \mathcal{D}} x_I x_J = 1[I = J]$). Then we can expand $p_0 : \{\pm 1\}^d \to \mathbb{R}$ in the Fourier basis as

$$p_0(x) = \sum (\bar{p}_0)_I x_I(x), \qquad (\bar{p}_0)_I = \mathbb{E}_{x \sim \mathcal{D}} p_0(x) x_I(x).$$

As $p_0$ is a probability function, there is $(\bar{p}_0)_\emptyset = 2^{-d}$. The set $P_S = \{(\bar{p}_0)_I : I \subset S\}$ gives the marginal distribution of $(x_i : i \in S)$. For instance, when $(\bar{p}_0)_{\{i,j\}} = (\bar{p}_0)_{\{i\}}(\bar{p}_0)_{\{j\}}$, $x_i$ and $x_j$ are independent in their marginal distribution.

#### 5.1.2 DIFFUSION MODELS

We adopt the conventions for diffusion models from Ho et al. (2020). Let $p_0$ be the distribution density we wish to sample from. While direct sampling from $p_0$ is computationally hard, We define a forward process $x_t$ where the signal gradually shrinks and Gaussian noise is added at each timestep for total $T$ steps.

$$x_0 \sim p_0(x), \quad p(x_t \mid x_{t-1}) = \mathcal{N}(\sqrt{1 - \beta_t} x_{t-1}, \beta_t \boldsymbol{I}), \ t = 1, 2, \ldots, T. \tag{3}$$

Here $0 < \beta_t < 1$ is a scale schedule of adding noise. Denote the distribution of $x_t$ as $p_t = \sqrt{\bar{\alpha}_t} p_0 * \mathcal{N}(0, (1 - \bar{\alpha}_t)\boldsymbol{I})$, $\alpha_t = 1 - \beta_t$, $\bar{\alpha}_t = \prod_{t=1}^{T} \alpha_t$. We choose the schedule so that $\bar{\alpha}_t \to 0$ as $t \to T$, so $p_t \to \mathcal{N}(0, \boldsymbol{I})$. If we can undo the forward noising process, then we can sample from $p_0$ by applying the reverse process on the samples from $p_T = \mathcal{N}(0, \boldsymbol{I})$, which is more computationally efficient. The reverse process is well-approximated by the following (Ho et al., 2020).

$$x_T \sim \mathcal{N}(0, \boldsymbol{I}), \quad x_{t-1} \sim \mathcal{N}\left(\frac{1}{\sqrt{\alpha_t}}\left(x_t - \frac{1 - \alpha_t}{\sqrt{1 - \bar{\alpha}_t}} s(x_t, t)\right), \tilde{\beta}_t \boldsymbol{I}\right) \tag{4}$$

where $s(x_t, t) = \frac{x_t - \sqrt{\bar{\alpha}_t} \mathbb{E}(x_0 | x_t)}{\sqrt{1 - \bar{\alpha}_t}}$ predicts the backwards direction.

**Score Learning Setting.** The backwards direction $s(x_t, t)$ requires the computation of the term $\mathbb{E}(x_0 | x_t)$ which may not be tractable in practice. Instead, we approximate the direction with a neural

network $s_\theta(x_t, t)$. For any $t > 0$, the training data is drawn from $p_t$ as $x_t = \sqrt{\bar\alpha_t} \cdot x_0 + \sqrt{1 - \bar\alpha_t} \cdot \xi$, where $x_0 \sim p_0$ and $\xi \sim \mathcal{N}(0, I)$, and the network is trained by minimizing

$$\mathcal{L}_t(\theta) = \mathbb{E}_{x_0, \xi} \|\xi - s_\theta(x_t, t)\|^2 = \mathbb{E}_{x_0, \xi} \left\| \frac{x_t - \sqrt{\bar\alpha_t} x_0}{\sqrt{1 - \bar\alpha_t}} - s_\theta(x_t, t) \right\|^2$$

$$= \mathbb{E}_{x_0, \xi} \left\| \frac{x_t - \sqrt{\bar\alpha_t} \mathbb{E}(x_0|x_t)}{\sqrt{1 - \bar\alpha_t}} - s_\theta(x_t, t) \right\|^2 + \mathbb{E}_{x_0, \xi} \left\| \frac{\sqrt{\bar\alpha_t}}{\sqrt{1 - \bar\alpha_t}} (x_0 - \mathbb{E}(x_0|x_t)) \right\|^2$$

Therefore the loss can be viewed as the square loss on the target score vector $y_t(x) = \frac{x_t - \sqrt{\bar\alpha_t} \mathbb{E}(x_0|x_t)}{\sqrt{1 - \bar\alpha_t}}$. In practices like natural images generation where $p_0$ is approximated by the empirical distribution of the training dataset, if the network $s_\theta(x, t)$ overfits by recovering the empirical target, then the diffusion model reproduces the training dataset. In this way the model memorizes the training data but cannot generate anything outside the training set. While ideal generation require the network to generalize properly, improper generalization causes hallucinatory samples. In the remaining part of the section, we present an example of such a bad generalization, which is manifested by the discrepancy of the LDR values.

## 5.2 LDR OF AN ACTUAL NEURAL NETWORK

In the following theoretical analysis, we use a two-layer neural network of hidden dimension $m$ as our score network. The $i$-th entry of the network output is $s_\theta^{(i)}(x, t) = \sum_{j \in [m]} a_{i,j,t} \sigma(w_{i,j,t}^\top x + b_{i,j,t})$ where $\sigma(x) = \max(x, 0)$ is the ReLU function, and the parameters $\theta = (a_{i,j,t}, w_{i,j,t}, b_{i,j,t})$ are updated via gradient flow (GF) as $\frac{d}{ds} \theta_s = -\nabla_\theta \mathcal{L}_t(\theta_s)$.

Notice that for $t > 0$, $\mathcal{L}_t$ is a smooth function as the data $x$ has a smooth probability density function over the space. For a starting point $\theta_0$, we use $\Phi(\theta_0, s)$ to denote the endpoint $\theta_s$ of gradient flow at time $s$, which is unique given the smoothness of the loss function. We will show that starting from a small initialization, the gradient flow boosts the LDR of the network, crossing the ground-truth LDR, and drives the network parameters close to an invariant set of LDR 1. Thereby, the resulting diffusion model of high LDR value creates hallucinatory generations.

## 5.3 HIGH LDR IS HARD TO LOWER FOR OPTIMIZATION

The experimental evidence hints a pattern of grammar rules that complicate the training process. We summarize such a pattern in Assumption 5.1. By Fourier transformation on the hypercube, we can expand the target probability $p_0(x) = \sum_{I \subset [d]} \bar{p}_0(I) x_I$ where $x_I = \prod_{i \in I} x_i$ are the Fourier basis.

**Assumption 5.1.** The grammar rule $p_0$ satisfies, for any $i, j \in [d]$, $\bar{p}_0(i) = 0$ and $\bar{p}_0(i, j) = 0$.

This means that the marginal distribution for any digit in the sequence is uniform, and for any pair of digits is independent. An example of such distributions are valid sequences uniformly drawn from a parity rule (e.g. satisfies $\prod_{i \in I} x_i = 1$ for any $I$ that $|I| > 2$). When the pairwise correlation disappears, we observe that the network is no longer able to escape an invariant set that has LDR 1.

**Theorem 5.2.** *Under Assumption 5.1, let $M = \{\theta : a_{i,j,t}(I - e_i e_i^\top) w_{i,j,t} = 0\}$, then $M$ is an invariant set under gradient flow. Namely, from any $\theta \in M$, gradient flow $\Phi(\theta, t) \in M$, $\forall t > 0$. For any $\theta \in M$, there is $LDR(\theta, \mathcal{R}) = 1$ for any $\mathcal{R} \subset [d]$.*

We observe that for any $\theta \in M$, $s_\theta^{(i)}(x, t) = \sum_{j \in [m]} a_{i,j,t} \sigma(w_{i,j,t}^\top x + b_{i,j,t})$ is irrelevant to the dimensions of $x$ other than $x^{(i)}$, therefore it cannot represent the true target score for any distribution that involves correlations over multiple dimensions. Actually since the reverse process starts from a Gaussian distribution that enjoys independent entries on different dimensions, the denoised sample will always have independent entries. For instance, for the generation of the parity rule, the best model in $M$ can only generate the uniform distribution over all of the hypercube, giving a half chance of hallucination. Therefore it is favorable to avoid optimizing the model into the set $M$. However, we will show that driving towards $M$ is implicitly induced by running gradient flow with small initialization.

## 5.4 EARLY TRAINING IS BIASED TOWARDS HIGH LDR

A series of previous works (Woodworth et al., 2020; Jin et al., 2023) adopts small initialization to assist representation learning for MLP networks, as opposed to the kernel regime where the network representations barely changes. We adopt the idea to check how network's representation change in the early phase of training. As exhibited in Figure 6, the network parameter will bias towards a 1-sparse feature extraction that marks the invariant set $M$ (Theorem 5.2), which we will prove in theory in the remaining part of the section.

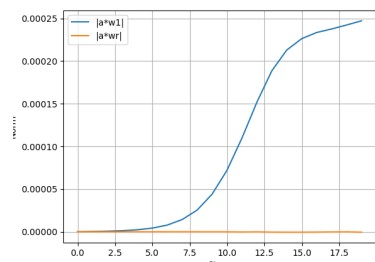

Figure 6: When learning $s_\theta^{(1)}(x_t, t)$, the average norm of neurons' $1_{st}$ dimension weight $|aw^{(1)}|$ increase much faster than $\|a(w - w^{(1)})\|$. It means $s_\theta^{(1)}(x_t, t)$ becomes a univariate function in $x_t^{(1)}$.

For a small constant $\sigma_{init}$ and constant $r > 0$, we use the initialization scheme for $\theta = (a_{i,j,t}, w_{i,j,t}, b_{i,j,t})$ as

$$w_{i,j,t}(0) \sim \mathcal{N}(0, \sigma_{init}^2), \quad b_{i,j,t}(0) \sim \mathcal{N}(0, \sigma_{init}^2 r^2),$$
$$a_{i,j,t}(0) \sim \text{Unif}(\{\pm 1\}) \sqrt{\|w_{i,j,t}(0)\|^2 + b_{i,j,t}(0)^2}.$$

Inspired by the G-function (Maennel et al., 2018; Lyu et al., 2021), when the initialization is very small, the neural network output $s_\theta(x, t) \approx 0$, so we can expand the loss as

$$\mathcal{L}_t(\theta) = \mathbb{E}_{x_t} \|s_\theta(x_t, t) - y_t(x_t)\|^2 = \mathbb{E}_{x_t}[\|y_t(x_t)\|^2 - 2s_\theta(x_t, t)^\top y_t(x_t)] + O(\mathbb{E}_{x_t}(s_\theta^2(x_t)))$$

So the initial trajectory of the neural network aims to optimize a surrogate loss $\tilde{\mathcal{L}}(\theta) = \mathbb{E}_{x_t}[-2s_\theta(x_t, t)^\top y_t(x_t)]$. For the simplicity of reasoning we will consider the trajectory $(\tilde{a}_{i,j,t}, \tilde{w}_{i,j,t}, \tilde{b}_{i,j,t})$ on such a surrogate loss, which can always well-approximate the early trajectory on the actual loss when $\sigma_{init}$ is small enough. We observe the following.

**Theorem 5.3.** *Under Assumption 5.1, for any $0 < c < 1$, there are real number $M_c, t_c$ that for a 2-layer ReLU network with width $m > M_c$, with high probability over the initialization, the network $\tilde{s}_{\tilde{\theta}(\tau)}(x, t) = \sum_{i \in [d], j \in [m]} \tilde{a}_{i,j,t} \sigma(\tilde{w}_{i,j,t}^\top x + \tilde{b}_{i,j,t}) e_i$ has high LDR for any time $\tau > t_c$ and any region $\mathcal{R} \subset [d]$. Specifically,*

$$LDR(\tilde{\theta}(\tau), \mathcal{R}) > 1 - c.$$

The proof of the theorem actually implies something stronger: not only does the network have LDR close to 1, its parameters go arbitrarily close to the set $M$. For statement conciseness we omit the details to Appendix A.2. The theorem depicts the representation learning process in the early phase of training: for a neuron along the direction $\tilde{w}_{i,j,t}(0) = \pm \|\tilde{w}_{i,j,t}(0)\| e_i$, with proper bias $\tilde{b}_{i,j,t}$, the growth rate of its norm (or $\tilde{a}_{i,j,t}$) is maximal and its direction does not alter, and therefore after a fixed training period it will have an exponentially larger impact to the network output than a neuron of suboptimal direction. Therefore after a few epochs, a neuron either still has a small magnitude ($|a_{i,j,t}|$), or its weight will be close to the optimal direction $\tilde{w}_{i,j,t}(s) \simeq e_i$, making the whole network close to the saddle set $M$ introduced in Theorem 5.2 and having high LDR. While it may take a long time for the network to escape the neighborhood of $M$, the network can operate inside $M$ to learn denoising functions that recovers each marginal distribution of $p_0$. In this way the model independently conducts denoising on each individual dimension, performing local generation that introduces hallucination in cases like the parity.

## 6 DISCUSSION

**Does local generation bias still hold for real world distribution?** In our analysis and synthetic text distribution, the condition that different token symbols are independent plays a critical role. Is our discovered bias and mechanism still robust when the distribution does not strictly satisfy this requirement? We conduct experiments on real world text distributions to answer these concerns. We construct two datasets, one rendering 1,000 common English words and the other contains 1,000 common Chinese characters. When using diffusion model to learn score matching and generate samples, we observe similar phenomenon persists to happen. Both models go through the "fuzzy-hallucination-overfitting" three phases, randomly assembling radicals and letters in nonsensical way

at intial stage, and overfit to duplicate training data after long period of training. We also test LDR in these scenarios, finding the same decreasing pattern. Note that we selcet $\sqrt{\bar{\alpha}_t} = 0.2$ because signal-noise-ratio increase significant at this stage and it is critical for determining the final content. Result shows the same local generation bias still exists at early stage of training and leads to hallucination for real text distribution. It also confirms that the decrease of LDR implies overfitting.

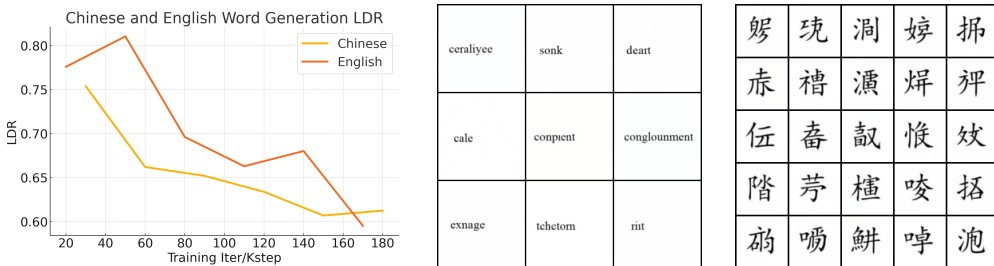

Figure 7: LDR trend for UNet learning Chinese and English texts (left). Hallucination example for misspelling words (middle) and Chinese characters (right).

Moreover, we also conduct the LDR analysis on real-world models such as FLUX1 (Labs, 2024) and StableDiffusion 3.5 (Podell et al., 2023). Since exact calculation the Jacobian matrix $J_{R,\theta}(x)$ requires large memory consumption for these enormous models, we use zeroth order approximation to calculate LDR. We verify that it also exhibits high LDR in generating images with text content, which corroborates our finding. For more details please refer to the appendix.

## 7 CONCLUSION

In this paper, we have presented a detailed investigation into the phenomenon of hallucinations in generating text-related contents. By combining empirical observations with theoretical analysis, we have demonstrated it to be closely related to the implicit bias of denoising network called local-generation bias. We find that such bias is not a mere consequence of the model's architecture but rather an inherent property of the training dynamics driven by score matching. It is shown that the trained diffusion models, despite their global receptive field capabilities, tend to rely on local information during the denoising process, generating symbols independently without capturing the global structure. The key to form this bias is the near pairwise independence between marginal distributions for each token symbol. We further introduce the Local Dependency Ratio (LDR) as a novel metric to quantify the extent of this local generation bias and applied it to various diffusion models, showing that this bias emerges early in training and persists through extensive training phases, even in real-world distribution where independent condition does not strictly hold. This study may shed light on the mechanisms behind hallucinations in diffusion models, highlighting the importance of addressing local generation bias for more accurate and coherent generation in tasks requiring complex global structure understanding.

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

# A APPENDIX

## A.1 PROOF FOR THE SADDLE POINT THEOREM 5.2

For any $\theta \in M$, we can set $w_{i,j,t} = c_{i,j,t} e_i$ for any neuron that $a_{i,j,t} \neq 0$, where $c_{i,j,t} \in \mathbb{R}$ are scalars. Then the network outputs $s_\theta^{(i)}(x) = \sum a_{i,j,t} \sigma(c_{i,j,t} x^{(i)} + b_{i,j,t})$ depend only on the $i$-th coordinate of the input $x$.

Now we compute the gradient for $w_{i,j,t}$ as

$$
\begin{aligned}
\frac{d}{ds} w_{i,j,t} &= -2\mathbb{E}_{x_t}(s_\theta^{(i)}(x_t) - y_i(x_t)) a_{i,j,t} \sigma'(w_{i,j,t}^\top x_t + b_{i,j,t}) x_t \\
&= -2\mathbb{E}_{x_0,\xi} s_\theta^{(i)}(x_t) a_{i,j,t} \sigma'(c_{i,j,t} x_t^{(i)} + b_{i,j,t}) x_t \\
&\quad + 2\mathbb{E}_{x_0,\xi} a_{i,j,t} \xi^{(i)} \sigma'(\sqrt{\bar\alpha_t} c_{i,j,t} x_0^{(i)} + \sqrt{1-\bar\alpha_t} c_{i,j,t} \xi^{(i)} + b_{i,j,t}) x_t
\end{aligned}
$$

Now we wish to show that $\frac{d}{ds} w_{i,j,t}$ is still along the direction of $e_i$, thereby gradient flow does not escape the region of $M$. This is done by the following two lemmas.

**Lemma A.1.** *For any $k \neq i$,*

$$
\mathbb{E}_{x_0,\xi} s_\theta^{(i)}(x_t) a_{i,j,t} \sigma'(c_{i,j,t} x_t^{(i)} + b_{i,j,t}) e_k^\top x_t = 0.
$$

*Proof.*

$$
\begin{aligned}
&\mathbb{E}_{x_0,\xi} s_\theta^{(i)}(x_t) \sigma'(c_{i,j,t} x_t^{(i)} + b_{i,j,t}) e_k^\top x_t \\
&= \mathbb{E}_{x_0,\xi} \sum_{j'} a_{i,j',t} \sigma(c_{i,j',t} x_t^{(i)} + b_{i,j',t}) \sigma'(c_{i,j,t} x_t^{(i)} + b_{i,j,t}) x_t^{(k)}
\end{aligned}
$$

Notice that by Assumption 5.1, $x_0^{(i)}$ and $x_0^{(k)}$ are independent, and for standard Gaussian $\xi^{(i)}$ and $\xi^{(k)}$ are independent. Furthermore $\mathbb{E}x_t^{(k)} = \sqrt{\bar{\alpha}_t}\mathbb{E}x_0^{(k)} + \sqrt{1-\bar{\alpha}_t}\mathbb{E}\xi^{(k)} = 0$. Therefore,

$$
\begin{aligned}
&\mathbb{E}_{x_0,\xi} s_\theta^{(i)}(x_t)\sigma'(c_{i,j,t}x_t^{(i)} + b_{i,j,t})e_k^\top x_t \\
&= \sum_{j'} a_{i,j',t}[\mathbb{E}_{x_t^{(i)}}\sigma(c_{i,j',t}x_t^{(i)} + b_{i,j',t})\sigma'(c_{i,j,t}x_t^{(i)} + b_{i,j,t})][\mathbb{E}_{x_t^{(k)}}x_t^{(k)}] \\
&= 0.
\end{aligned}
$$

$\square$

**Lemma A.2.** *For any $k \neq i$,*

$$
\mathbb{E}_{x_0,\xi}\xi^{(i)}\sigma'(\sqrt{\bar{\alpha}_t}c_{i,j,t}x_0^{(i)} + \sqrt{1-\bar{\alpha}_t}c_{i,j,t}\xi^{(i)} + b_{i,j,t})e_k^\top x_t = 0.
$$

*Proof.* Similarly, $x_t^{(k)} = \sqrt{\bar{\alpha}_t}x_0^{(k)} + \sqrt{1-\bar{\alpha}_t}\xi^{(k)}$ has mean 0. Since $(x_0^{(k)}, \xi^{(k)})$ and $(x_0^{(i)}, \xi^{(i)})$ are independent, we know

$$
\begin{aligned}
&\mathbb{E}_{x_0,\xi}\xi^{(i)}\sigma'(\sqrt{\bar{\alpha}_t}c_{i,j,t}x_0^{(i)} + \sqrt{1-\bar{\alpha}_t}c_{i,j,t}\xi^{(i)} + b_{i,j,t})e_k^\top x_t \\
&= [\mathbb{E}_{x_0^{(i)},\xi^{(i)}}\xi^{(i)}\sigma'(\sqrt{\bar{\alpha}_t}c_{i,j,t}x_0^{(i)} + \sqrt{1-\bar{\alpha}_t}c_{i,j,t}\xi^{(i)} + b_{i,j,t})][\mathbb{E}_{x_t^{(k)}}x_t^{(k)}] \\
&= 0.
\end{aligned}
$$

$\square$

Besides, for a neuron that $a_{i,j,t} = 0$, from Lemma A.3 we know $w_{i,j,t} = 0$ and $b_{i,j,t} = 0$, so $\frac{d}{ds}a_{i,j,t} = 0$. So $a_{i,j,t}$ will keep zero along the trajectory. Thus $M$ is indeed an invariant set under gradient flow.

Finally we calculate the LDR value for the network. Notice that

$$
\frac{\partial}{\partial x}s_\theta^{(i)}(x) = \sum a_{i,j,t}\sigma'(c_{i,j,t}x^{(i)} + b_{i,j,t})c_{i,j,t}e_i
$$

is always along the direction $e_i$, for any $\mathcal{R} \subset [d]$ and $j \notin \mathcal{R}$ there is

$$
(\frac{\partial}{\partial x}s_\theta^{(\mathcal{R})}(x))^\top e_j = 0_{|\mathcal{R}|}.
$$

Therefore by definition there is $LDR(\theta, \mathcal{R}) = 1$.

### A.2 PROOF FOR THE IMPLICIT TRAINING BIAS THEOREM 5.3

Here we consider a fixed $t$ and target dimension $i$, and we omit the subscripts of $t$ and $i$ for the simplicity of notations. Thus for the network $s_\theta(x) = \sum_{j \in [m]} a_j\sigma(w_j^\top x + b_j)$, we optimize it via GF on the square loss $\mathcal{L}(\theta) = \mathbb{E}_{x_t=x}(s_\theta(x) - y_i(x))^2$ as

$$
\begin{aligned}
\frac{d}{ds}a_j &= -2\mathbb{E}_x(s_\theta(x) - y_i(x))\sigma(w_j^\top x + b_j) \\
\frac{d}{ds}w_j &= -2\mathbb{E}_x(s_\theta(x) - y_i(x))a_j\sigma'(w_j^\top x + b_j)x \\
\frac{d}{ds}b_j &= -2\mathbb{E}_x(s_\theta(x) - y_i(x))a_j\sigma'(w_j^\top x + b_j)
\end{aligned}
$$

We write $\theta(s)$ to denote the value of the parameters at time $s$. First we observe that the two layers of the network stay balanced throughout the course of the training process.

**Lemma A.3.** $\frac{d}{ds}(a_j^2 - \|w_j\|^2 - b_j^2) = 0$.

*Proof.* This is obtained directly as

$$\frac{d}{ds}(a_j^2 - \|w_j\|^2 - b_j^2) = 2a_j\frac{d}{ds}a_j - 2w_j^\top\frac{d}{ds}w_j - 2b_j\frac{d}{ds}b_j$$
$$= -4\mathbb{E}_x(s_\theta(x) - y(x))a_j\left[\sigma(w_j^\top x + b_j)\right.$$
$$\left. - \sigma'(w_j^\top x + b_j)(w_j^\top x + b_j)\right]$$
$$= 0.$$

$\square$

Therefore $a_j = \mathrm{sgn}(a_j)\sqrt{\|w_j\|^2 + b_j^2}$ through out the process.

### A.2.1 Growth Rate For the First Layer Weight

Inspired by the G-function (Maennel et al., 2018; Lyu et al., 2021), when the initialization is very small, the neural network output $s_\theta(x) \approx 0$, so we can expand the loss as

$$\mathcal{L}(\theta) = \mathbb{E}_x(s_\theta(x) - y_i(x))^2 = \mathbb{E}_x y_i(x)^2 - 2s_\theta(x)y_i(x) + O(s_\theta^2(x))$$

So the initial trajectory of the neural network aims to optimize a surrogate loss $\tilde{\mathcal{L}}(\theta) = \mathbb{E}_x - 2s_\theta(x)y_i(x)$. We define the parameters $\tilde{\theta} = (\tilde{a}_j, \tilde{w}_j, \tilde{b}_j)$ to be the parameters run specifically for the surrogate loss, namely, let $\tilde{\theta}(0) = \theta(0)$ and

$$\frac{d}{ds}\tilde{a}_j = 2\mathbb{E}_x y_i(x)\sigma(\tilde{w}_j^\top x + \tilde{b}_j)$$
$$\frac{d}{ds}\tilde{w}_j = 2\mathbb{E}_x y_i(x)\tilde{a}_j\sigma'(\tilde{w}_j^\top x + \tilde{b}_j)x$$
$$\frac{d}{ds}\tilde{b}_j = 2\mathbb{E}_x y_i(x)\tilde{a}_j\sigma'(\tilde{w}_j^\top x + \tilde{b}_j).$$

We will have similarly, $\frac{d}{ds}(\tilde{a}_j^2 - \|\tilde{w}_j\|^2 - \tilde{b}_j^2) = 0$, so $\tilde{a}_j = \mathrm{sgn}(\tilde{a}_j)\sqrt{\|\tilde{w}_j\|^2 + \tilde{b}_j^2}$ through out the process. Then we can actually show that the scale of $\tilde{a}_j$ grows exponentially as a function of the direction of $(\frac{\tilde{w}_j}{|\tilde{a}_j|}, \frac{\tilde{b}_j}{|\tilde{a}_j|})$ as

**Theorem A.4.** *Under Assumption 5.1, the weight of each neuron $|a_{i,j,t}|$ grows exponentially in time: for every $i,t$, there exists a function $K_{i,t} : S^d \to \mathbb{R}$ such that*

$$|\tilde{a}_{i,j,t}(s)| = |\tilde{a}_{i,j,t}(0)| \exp\left(2\mathrm{sgn}(\tilde{a}_{i,j,t}(0))\int_0^s K_{i,t}\left(\frac{\tilde{w}_{i,j,t}(\tau)}{|\tilde{a}_{i,j,t}|(\tau)}, \frac{\tilde{b}_{i,j,t}(\tau)}{|\tilde{a}_{i,j,t}|(\tau)}\right) d\tau\right)$$

*The function $K_{i,t}$ marks the growth rate. The rate satisfy*

- $0 < K_{i,t}(w,b) < \sqrt{1 - \bar{\alpha}_t}$ *when $w^{(i)} > 0$; $0 > K_{i,t}(w,b) > -\sqrt{1 - \bar{\alpha}_t}$, when $w^{(i)} < 0$.*

- *When $w^{(i)} > 0$, the maximal value of $K_{i,t}(w,b)$ is uniquely achieved at $(w,b) = \frac{1}{\sqrt{1+(D^*)^2}}(e_i, D^*)$ for some $D^* > 0$; When $w^{(i)} < 0$, the minimal value of $K_{i,t}(w,b)$ is uniquely achieved at $(w,b) = \frac{1}{\sqrt{1+(D^*)^2}}(-e_i, D^*)$.*

- *the maximally-growing neuron directions $\left(\frac{\tilde{w}_{i,j,t}(\tau)}{|\tilde{a}_{i,j,t}|(\tau)}, \frac{\tilde{b}_{i,j,t}(\tau)}{|\tilde{a}_{i,j,t}|(\tau)}\right) = \frac{1}{\sqrt{1+(D^*)^2}}(e_i, D^*)$ for $a_{i,j,t} > 0$ and $\left(\frac{\tilde{w}_{i,j,t}(\tau)}{|\tilde{a}_{i,j,t}|(\tau)}, \frac{\tilde{b}_{i,j,t}(\tau)}{|\tilde{a}_{i,j,t}|(\tau)}\right) = \frac{1}{\sqrt{1+(D^*)^2}}(-e_i, D^*)$ for $a_{i,j,t} < 0$ are invariant under gradient flow.*

- $a_{i,j,t}\frac{d}{d\tau}K_{i,t}\left(\frac{\tilde{w}_{i,j,t}(\tau)}{|\tilde{a}_{i,j,t}|(\tau)}, \frac{\tilde{b}_{i,j,t}(\tau)}{|\tilde{a}_{i,j,t}|(\tau)}\right) \geq 0.$

We will break the theorem by parts. First let's define the function $K$ with the following lemma.

**Lemma A.5.** *There exists a function $K : S^d \to \mathbb{R}$ such that*

$$|\tilde{a}_j(s)| = |\tilde{a}_j(0)| \exp\left(2 \int_0^s K\left(\frac{\tilde{w}_j(\tau)}{|\tilde{a}_j|(\tau)}, \frac{\tilde{b}_j(\tau)}{|\tilde{a}_j|(\tau)}\right) d\tau\right)$$

*when $\tilde{a}_j(0) > 0$, and*

$$|\tilde{a}_j(s)| = |\tilde{a}_j(0)| \exp\left(-2 \int_0^s K\left(\frac{\tilde{w}_j(\tau)}{|\tilde{a}_j|(\tau)}, \frac{\tilde{b}_j(\tau)}{|\tilde{a}_j|(\tau)}\right) d\tau\right)$$

*when $\tilde{a}_j(0) < 0$.*

*Proof.* We know

$$\frac{d}{ds}\tilde{a}_j = 2\mathbb{E}_x y_i(x)\sigma(\tilde{w}_j^\top x + \tilde{b}_j)$$

$$= 2|\tilde{a}_j|\mathbb{E}_x y_i(x)\sigma\left(\frac{\tilde{w}_j}{|\tilde{a}_j|}^\top x + \frac{\tilde{b}_j}{|\tilde{a}_j|}\right)$$

The proof is then done by taking $K(w, b) = \mathbb{E}_x y_i(x)\sigma(w^\top x + b)$. Notice that we only query $K$ when $\|w\|^2 + b^2 = 1$. □

*Proof of Theorem A.4.* Now we take a closer examination of the function $K$. Let $e_i$ be the unit vector along the $i$-th dimension and $P_i = I - e_i e_i^\top$ be the projection matrix removing the $i$-th dimension. Since the data $x$ is sampled through the process $x = \sqrt{\bar{\alpha}_t}x_0 + \sqrt{1 - \bar{\alpha}_t}\xi$ for $x_0 \in \{\pm 1\}^d$ and $\xi \sim \mathcal{N}(0, I)$, we know

$$K(w, b) = \mathbb{E}_x y_i(x)\sigma(w^\top x + b)$$
$$= \mathbb{E}_{x_0, \xi}(\xi^{(i)}\sigma(\sqrt{\bar{\alpha}_t}w^\top x_0 + \sqrt{1 - \bar{\alpha}_t}w^\top P_i\xi + \sqrt{1 - \bar{\alpha}_t}w^{(i)}\xi^{(i)} + b))$$

Define $A = \sqrt{\bar{\alpha}_t}w^\top x_0 + \sqrt{1 - \bar{\alpha}_t}w^\top P_i\xi + b$, $B = \sqrt{1 - \bar{\alpha}_t}w^{(i)}$. since both $A, B$ are independent to $\xi^{(i)}$,

$$K(w, b) = \mathbb{E}_{x_0, \xi}\xi^{(i)}\sigma(A + B\xi^{(i)})$$
$$= \frac{1}{2}\mathbb{E}_A\left(B + |B|\text{erf}\left(\frac{A}{\sqrt{2}B}\right)\right)$$

where we use the standard error function as $\text{erf}(x) = \frac{2}{\sqrt{\pi}}\int_0^x e^{-s^2}ds \in [-1, 1]$. Furthermore, define $C = (\sqrt{\bar{\alpha}_t}w^\top P_i x_0 + \sqrt{1 - \bar{\alpha}_t}w^\top P_i\xi + b)^2 + (w^{(i)})^2$, $D = \frac{\sqrt{\bar{\alpha}_t}w^\top P_i x_0 + \sqrt{1 - \bar{\alpha}_t}w^\top P_i\xi + b}{w^{(i)}}$, we know $B = \text{sgn}(B)\sqrt{(1 - \bar{\alpha}_t)\frac{C}{1+D^2}}$, and

$$K(w, b) = \frac{\sqrt{1 - \bar{\alpha}_t}}{2}\left(\mathbb{E}\text{sgn}(B)\sqrt{\frac{C}{1+D^2}} + \mathbb{E}_{x_0^{(i)}=1}\frac{1}{2}\sqrt{\frac{C}{1+D^2}}\text{erf}\left(\frac{D}{\sqrt{2(1-\bar{\alpha}_t)}} + \sqrt{\frac{\bar{\alpha}_t}{2(1-\bar{\alpha}_t)}}\right)\right.$$

$$\left. + \mathbb{E}_{x_0^{(i)}=-1}\frac{1}{2}\sqrt{\frac{C}{1+D^2}}\text{erf}\left(\frac{D}{\sqrt{2(1-\bar{\alpha}_t)}} - \sqrt{\frac{\bar{\alpha}_t}{2(1-\bar{\alpha}_t)}}\right)\right)$$

Observe that as $x_0$ and $\xi$ are independent with $\mathbb{E}x_0 = \mathbb{E}\xi = 0$, and $x_0$ have pairwise independent entries,

$$\mathbb{E}C|x_0^{(i)} = \mathbb{E}\bar{\alpha}_t(w^\top P_i x_0)^2 + (1 - \bar{\alpha}_t)(w^\top P_i\xi)^2 + (w^{(i)})^2 + b^2 = \|w\|^2 + b^2 = 1.$$

When $w^{(i)} > 0$, since $\text{erf}(x) \in [-1, 1]$, we always have $K(w, b) > 0$. In this case, by Jensen's inequality,

$$K(w, b) \le \frac{\sqrt{1 - \bar{\alpha}_t}}{2}\sup_{D \in \mathbb{R}}\frac{1}{\sqrt{1+D^2}}\left(1 + \frac{1}{2}\text{erf}\left(\frac{D}{\sqrt{2(1-\bar{\alpha}_t)}} + \sqrt{\frac{\bar{\alpha}_t}{2(1-\bar{\alpha}_t)}}\right)\right.$$

$$\left. + \frac{1}{2}\text{erf}\left(\frac{D}{\sqrt{2(1-\bar{\alpha}_t)}} - \sqrt{\frac{\bar{\alpha}_t}{2(1-\bar{\alpha}_t)}}\right)\right)$$

Symmetrically as erf is an odd function, when $w^{(i)} < 0$, there is $K(w, b) < 0$, and

$$K(w, b) \geq -\frac{\sqrt{1 - \bar{\alpha}_t}}{2} \sup_{D \in \mathbb{R}} \frac{1}{\sqrt{1 + D^2}} \left(1 - \frac{1}{2}\mathrm{erf}\left(\frac{D}{\sqrt{2(1 - \bar{\alpha}_t)}} + \sqrt{\frac{\bar{\alpha}_t}{2(1 - \bar{\alpha}_t)}}\right)\right.$$
$$\left. -\frac{1}{2}\mathrm{erf}\left(\frac{D}{\sqrt{2(1 - \bar{\alpha}_t)}} - \sqrt{\frac{\bar{\alpha}_t}{2(1 - \bar{\alpha}_t)}}\right)\right).$$

Since erf is odd, $K(w, b) = -K(-w, b)$, so we only need to consider the case where $w^{(i)} > 0$. In this case, the maximum of $|K|$ is achieved when $\mathbb{E}C = (\mathbb{E}\sqrt{C})^2$ and $D = D^* > 0$ that maximizes the above functions, namely when $w^\top P_i = 0$ and $\frac{b}{|w^{(i)}|} = D^*$. By the first-order condition of optimality, let

$$f(D) = \frac{1}{\sqrt{1 + D^2}} \left(1 + \frac{1}{2}\mathrm{erf}\left(\frac{D}{\sqrt{2(1 - \bar{\alpha}_t)}} + \sqrt{\frac{\bar{\alpha}_t}{2(1 - \bar{\alpha}_t)}}\right) + \frac{1}{2}\mathrm{erf}\left(\frac{D}{\sqrt{2(1 - \bar{\alpha}_t)}} - \sqrt{\frac{\bar{\alpha}_t}{2(1 - \bar{\alpha}_t)}}\right)\right)$$

(5)

then we know $\frac{d}{dD}f(D^*) = 0$, namely

$$\frac{1 + (D^*)^2}{D^*\sqrt{2\pi(1 - \bar{\alpha}_t)}} \left(\exp\left[-\left(\frac{D^*}{\sqrt{2(1 - \bar{\alpha}_t)}} + \sqrt{\frac{\bar{\alpha}_t}{2(1 - \bar{\alpha}_t)}}\right)^2\right] + \exp\left[-\left(\frac{D^*}{\sqrt{2(1 - \bar{\alpha}_t)}} - \sqrt{\frac{\bar{\alpha}_t}{2(1 - \bar{\alpha}_t)}}\right)^2\right]\right)$$
$$= \left(1 + \frac{1}{2}\mathrm{erf}\left(\frac{D^*}{\sqrt{2(1 - \bar{\alpha}_t)}} + \sqrt{\frac{\bar{\alpha}_t}{2(1 - \bar{\alpha}_t)}}\right) + \frac{1}{2}\mathrm{erf}\left(\frac{D^*}{\sqrt{2(1 - \bar{\alpha}_t)}} - \sqrt{\frac{\bar{\alpha}_t}{2(1 - \bar{\alpha}_t)}}\right)\right)$$

**Lemma A.6.** $|K(w, b)| \leq \sqrt{1 - \bar{\alpha}_t}.$

*Proof.* By symmetry, WLOG we consider the case where $w^{(i)} > 0$. Then

$$K(w, b) \leq \frac{\sqrt{1 - \bar{\alpha}_t}}{2} \sup_{D \in \mathbb{R}} \frac{1}{\sqrt{1 + D^2}} \left(1 + \frac{1}{2}\mathrm{erf}\left(\frac{D}{\sqrt{2(1 - \bar{\alpha}_t)}} + \sqrt{\frac{\bar{\alpha}_t}{2(1 - \bar{\alpha}_t)}}\right)\right.$$
$$\left. +\frac{1}{2}\mathrm{erf}\left(\frac{D}{\sqrt{2(1 - \bar{\alpha}_t)}} - \sqrt{\frac{\bar{\alpha}_t}{2(1 - \bar{\alpha}_t)}}\right)\right)$$
$$\leq \frac{\sqrt{1 - \bar{\alpha}_t}}{2}(1 + \frac{1}{2} + \frac{1}{2})$$
$$= \sqrt{1 - \bar{\alpha}_t}.$$

$\square$

Next we examine the dynamics of a neuron along this optimal direction $\left(\frac{\tilde{w}_j(\tau)}{|\tilde{a}_j|(\tau)}, \frac{\tilde{b}_j(\tau)}{|\tilde{a}_j|(\tau)}\right) = \frac{1}{\sqrt{1 + (D^*)^2}}(e_i, D^*)$ with $w^\top P_i = 0$ and $\frac{b}{w^{(i)}} = D^*$. By Theorem 5.2 we know that the gradient of $\tilde{w}_j$ is also along the direction of $e_i$; furthermore, direct calculation plugging the above first-order condition, we arrive at

$$\frac{\frac{d}{ds}b}{\frac{d}{ds}w^{(i)}} = \frac{\mathbb{E}_{x_0,\xi}\xi\sigma'((\sqrt{\bar{\alpha}_t}x_0 + \sqrt{1 - \bar{\alpha}_t}\xi + D^*)w^{(i)})}{\mathbb{E}_{x_0,\xi}\xi(\sqrt{\bar{\alpha}_t}x_0 + \sqrt{1 - \bar{\alpha}_t}\xi)\sigma'((\sqrt{\bar{\alpha}_t}x_0 + \sqrt{1 - \bar{\alpha}_t}\xi + D^*)w^{(i)})} = D^*.$$

This shows that a neuron along the direction $\left(\frac{\tilde{w}_j(\tau)}{|\tilde{a}_j|(\tau)}, \frac{\tilde{b}_j(\tau)}{|\tilde{a}_j|(\tau)}\right) = (w, b) = \frac{1}{\sqrt{1 + (D^*)^2}}(e_i, D^*)$ keeps the same direction during the course of the dynamics, thus the weight $\tilde{a}_j$ can maintain the maximum growing rate.

Finally we calculate the rate of change of the function $K$. Since for the vector $z(\tau) = (\tilde{w}_j(\tau), \tilde{b}_j(\tau))$, $\frac{z(\tau)}{\|z(\tau)\|} = (\frac{\tilde{w}_j(\tau)}{|\tilde{a}_j|(\tau)}, \frac{\tilde{b}_j(\tau)}{|\tilde{a}_j|(\tau)}) \in S^d$, we know

$$\frac{d}{d\tau}\frac{z(\tau)}{\|z(\tau)\|} = (I - \frac{zz^\top}{z^\top z}(\tau))\frac{1}{\|z(\tau)\|}\frac{d}{d\tau}z(\tau)$$

$$= (I - \frac{zz^\top}{z^\top z})\frac{1}{|\tilde{a}_j|}2\mathbb{E}_x y_i(x)\tilde{a}_j\sigma'(\tilde{w}_j^\top x + \tilde{b}_j)(x, 1).$$

Meanwhile, as we calculate $\nabla K$ in the tangent space of $z/\|z\|$ on $S^d$,

$$\nabla K(\frac{z(\tau)}{\|z(\tau)\|}) = (I - \frac{zz^\top}{z^\top z})\mathbb{E}_x y_i(x)\sigma'((\frac{\tilde{w}_j}{|\tilde{a}_j|})^\top x + \frac{\tilde{b}_j}{|\tilde{a}_j|})(x, 1)$$

$$= (I - \frac{zz^\top}{z^\top z})\mathbb{E}_x y_i(x)\sigma'(\tilde{w}_j^\top x + \tilde{b}_j)(x, 1)$$

Therefore

$$\tilde{a}_j(\tau)\frac{d}{d\tau}K\left(\frac{z(\tau)}{\|z(\tau)\|}\right) = \tilde{a}_j(\tau)\nabla K\left(\frac{z(\tau)}{\|z(\tau)\|}\right)\frac{d}{d\tau}\frac{z(\tau)}{\|z(\tau)\|}$$

$$= 2|\tilde{a}_j(\tau)|\left\|(I - \frac{zz^\top}{z^\top z})\mathbb{E}_x y_i(x)\sigma'(\tilde{w}_j^\top x + \tilde{b}_j)(x, 1)\right\|^2 \geq 0.$$

$\square$

### A.2.2 THE LDR DYNAMICS

Here we provide a proof for the main theorem Theorem 5.3. First we give a characterization of the LDR for models near the invariant set $M$.

**Lemma A.7.** *If a network $s_\theta^{(i)}(x) = \sum_j a_j\sigma(w_j^T x + b_j)$ has for $k_1 > 0$, $k_2 > k_0$, $1 > k_4 > k_3 > 0$,*

- *For all $j$, $a_j^2 = \|w_j\|^2 + b_j^2$.*

- *For all $j$, either $|a_j| \leq k_0$ or $|a_j|\|w_j - w_j^{(i)}e_i\| \leq k_1 a_j w_j^{(i)}$.*

- *There is $j, j'$ such that $a_j \geq k_2$, $a_{j'} \leq -k_2$, $\frac{b_j}{|w_j^{(i)}|}, \frac{b_{j'}}{|w_{j'}^{(i)}|} \in [k_3, k_4]$.*

*Then there is function $P$ such that the LDR for the network can be bounded as*

$$LDR(\theta, \{i\}) \geq P(k_1, k_3, k_4)\max(\frac{k_2^2 - mk_0^2\sqrt{1 + k_4^2}}{k_2^2(1 + k_1) + mk_0^2\sqrt{1 + k_4^2}}, 0)^2.$$

*Furthermore $P$ is continuous with $\lim_{k_1 to 0} P(k_1, k_3, k_4) = 1$.*

Namely, a network will have high LDR when

- any neuron $j$ either has small weight $|a_j|$ (thus does not contribute much to the network output) or its weight $w_i$ align with $e_i$ well;
- for the majority of data, at least one neuron with large weight is activated.

Specifically, the first condition states that the network parameter is close to some parameters in the invariant set $M$, and the second ensures that such a closeness is on a function level, so that the closeness in the parameter space can be converted into the closeness in the function space, and eventually into the closeness in the LDR measure which only depends on the functions.

*Proof.* We calculate

$$\frac{\partial}{\partial x}s_\theta(x) = \sum_j a_j\sigma'(w_j^\top x + b_j)w_j.$$

Therefore by the definition of LDR, we have for the numerator,

$$\|e_i^\top \frac{\partial}{\partial x} s_\theta(x)\|^2 = (\sum_j a_j \sigma'(w_j^\top x + b_j)w_j^{(i)})^2$$

$$\geq \max(\sum_{|a_j| \geq k_0} a_j \sigma'(w_j^\top x + b_j)w_j^{(i)} - mk_0^2, 0)^2,$$

and for the denominator,

$$\|\frac{\partial}{\partial x} s_\theta(x)\|^2 = \|\sum_j a_j \sigma'(w_j^\top x + b_j)w_j\|^2$$

$$\leq [\|\sum_{|a_j| \geq k_0} a_j \sigma'(w_j^\top x + b_j)w_j\| + mk_0^2]^2$$

$$\leq [(\sum_{|a_j| \geq k_0} a_j \sigma'(w_j^\top x + b_j)w_j^{(i)})(1 + k_1) + mk_0^2]^2$$

Furthermore there is

$$\sum_{|a_j| \geq k_0} a_j \sigma'(w_j^\top x + b_j)w_j^{(i)} \geq \frac{k_2^2}{\sqrt{1+k_4^2}}[\sigma'(w_j^\top x + b_j) + \sigma'(w_{j'}^\top x + b_{j'})]$$

Therefore by definition we have

$$LDR(\theta, \{i\}) \geq \Pr(w_j^\top x + b_j > 0 \vee w_{j'}^\top x + b_{j'} > 0) \max(\frac{k_2^2 - mk_0^2\sqrt{1+k_4^2}}{k_2^2(1+k_1) + mk_0^2\sqrt{1+k_4^2}}, 0)^2.$$

Let the function

$$P(k_1, k_3, k_4) = \inf\{\Pr(w^\top x + b > 0 \vee (w')^\top x + (b') > 0):$$
$$\|w\| = \|w'\| = 1, \|w - w^{(i)}e_i\| \leq k_1 w^{(i)}, \|w' - (w')^{(i)}e_i\| \leq -k_1(w')^{(i)},$$
$$b, b' \in [k_3, k_4]\}$$

Then we know $\Pr(w_j^\top x + b_j > 0 \vee w_{j'}^\top x + b_{j'} > 0) \geq P(k_1, k_3, k_4)$.

Finally, as $k_1 \to 0$, there is $\|w - w^{(i)}e_i\|, \|w' - (w')^{(i)}e_i\| \to 0$. Since $b, b', w^{(i)} > 0$ and $(w')^{(i)} < 0$, so the set

$$\{w^\top x + b > 0 \vee (w')^\top x + (b') > 0\} \to \{w^{(i)}x^{(i)} + b > 0 \vee (w')^{(i)}x^{(i)} + (b') > 0\} = \mathbb{R}^d.$$

By the continuity of the probability measure for $x$, we know the function $P$ has limit $\Pr(x \in \mathbb{R}^d) = 1$. □

*Proof of Theorem 5.3.* Since $LDR(\theta, \mathcal{R}_1 \cup \mathcal{R}_2) \geq LDR(\theta, \mathcal{R}_1) + LDR(\theta, \mathcal{R}_2)$, WLOG we consider the case that $\mathcal{R} = \{i\}$ to be of size 1.

For any $c > 0$, let $K_0 = \max(\sup\{K(w, b) : \|w - w^{(i)}e_i\| > \frac{c}{8}w^{(i)}\}, \frac{\sqrt{1-\bar{\alpha}_t}}{2})$. From the discussion of the function $K$ from Theorem A.4, we know $0 < K_0 < \sqrt{1 - \bar{\alpha}_t}$. Since the function $K$ has a unique maximum at $\frac{1}{\sqrt{1+(D^*)^2}}(e_i, D^*)$, we can choose a real number $\delta \in (0, \sqrt{1 - \tilde{\alpha}_t} - K_0)$ and a neighborhood $O_\epsilon = \{(w, b) : \|w - w^{(i)}e_i\| < \epsilon w^{(i)}, |\frac{b}{w^{(i)}} - D^*| < \epsilon\}$ such that for all $(w, b) \notin O_\epsilon$, $K(w, b) \leq \sqrt{1 - \bar{\alpha}_t} - \delta$. Pick $k_3 = D^* - \epsilon$, $k_4 = D^* + \epsilon$, and $k_1 < \frac{c}{8}$ such that $P(k_1, k_3, k_4) > 1 - \frac{c}{2}$ in Lemma A.7.

Now pick $M_c$ so that with high probability, at initialization there are neurons $j$, $j'$ such that $a_j > 0$, $K(\frac{w_j}{a_j}, \frac{b_j}{a_j}) > \sqrt{1 - \tilde{\alpha}_t} - \delta$ and $a_{j'} < 0$, $K(\frac{w_{j'}}{|a_{j'}|}, \frac{b_{j'}}{|a_{j'}|}) < -\sqrt{1 - \tilde{\alpha}_t} + \delta$. Since the neuron parameters follow i.i.d. initial distributions, this is always possible with enough network width.

From Theorem A.4, we have the following facts:

- $\tilde{a}_j(\tau) \geq \tilde{a}_j(0)e^{2\tau(\sqrt{1-\bar{\alpha}_t}-\delta)}; \tilde{a}_{j'}(\tau) \leq \tilde{a}_{j'}(0)e^{2\tau(\sqrt{1-\bar{\alpha}_t}-\delta)}$.

- For any neuron $k$, if $\operatorname{sgn}(\tilde{a}_k(\tau))K(\frac{\tilde{w}_k(\tau)}{|\tilde{a}_k|(\tau)}, \frac{\tilde{b}_k(\tau)}{|\tilde{a}_k|(\tau)}) > K_0$, from the definition of $K_0$ there must be $\|\tilde{w}_k(\tau) - \tilde{w}_k^{(i)}(\tau)e_i\| \leq \frac{c}{8}|\tilde{w}_k^{(i)}(\tau)|$; otherwise for all $0 \leq s \leq \tau$ there is $\operatorname{sgn}(\tilde{a}_k(s))K(\frac{\tilde{w}_k(s)}{|\tilde{a}_k|(s)}, \frac{\tilde{b}_k(s)}{|\tilde{a}_k|(s)}) \leq \operatorname{sgn}(\tilde{a}_k(\tau))K(\frac{\tilde{w}_k(\tau)}{|\tilde{a}_k|(\tau)}, \frac{\tilde{b}_k(\tau)}{|\tilde{a}_k|(\tau)}) \leq K_0$, so we will have
$$|\tilde{a}_j(\tau)| \leq |\tilde{a}_j(0)|e^{2\tau K_0}.$$

Now pick $\tau_c$ so that $\tau_c > \frac{1}{2(\sqrt{1-\bar{\alpha}_t}-\delta-K_0)}\ln\frac{64m\sup_k|\tilde{a}_k(0)|}{\min(|a_j(0)|,|a_{j'}(0)|)(4c+c^2)\sqrt{1+k_4^2}}$, then we can apply Lemma A.7 with $k_0 = \sup_k|\tilde{a}_k(0)|e^{2\tau K_0}$ and $k_2 = \min(|a_j(0)|,|a_{j'}(0)|)e^{2\tau(\sqrt{1-\bar{\alpha}_t}-\delta)}$, then the LDR score for $\tau > \tau_c$ is

$$LDR(\theta, \{i\}) \geq (1 - \frac{c}{2})(\frac{1 - \frac{c}{16} - \frac{c^2}{64}}{1 + \frac{3c}{16} + \frac{c^2}{64}})^2 \geq (1 - \frac{c}{2})(1 - \frac{c}{4})^2 > 1 - c.$$

$\square$

### A.3 EXPERIMENTAL DETAILS

The details of experiment formulation is as below. Recall that a text distribution includes a set of discrete symbols $\mathcal{S} = \{s_1, s_2, \ldots, s_K\}$ and a spelling/grammar rule $P_G$. A list of symbol tokens are further rendered into ambient space by a function $h : \mathcal{S} \mapsto \mathbb{R}^d$ which maps each symbol to a vector in ambient space like image pixels or a single scalar. The full signal is obtained by concatenating these vectors. We describe $\mathcal{S}$ and $P_G$ we used in experiments.

**Parity:** There are only two symbols $\mathcal{S} = \{1, -1\}$. The rule is that there needs to be even number of symbol $s_1$. Namely $P_G(\mathcal{I}) = \frac{1}{2^{L-1}} \cdot \mathbb{I}\left[\prod_{j=1}^{L} s_{i_j} = 1\right]$. The ambient space rendering function can either by a single scalar $h(s_i) = s_i$. It can also be two pixel image or embedding vector templates in ambient observation space $\{o_{-1}, o_1\}$ and $h(s_i) = o_{s_i}$.

**Quarter-MNIST:** We combine four MNIST digits' image to become a whole figure. the symbol system is all digits $\mathcal{S} = [9]$. We fix the length $L = 4$ and requires that $s_1 + s_2 = s_3 + s_4$, and we have $P_G(\mathcal{I}) = \frac{1}{Z} \cdot \mathbb{I}[s_1 + s_2 = s_3 + s_4]$, $Z = 670$ is some normalization constant. The ambient space rendering function is a probabilistic image drawing function $h : \{0, \ldots, 9\} \mapsto \mathbb{R}^d$ which maps each digit to its hand-writing image.

**Dyck:** We also test dyck grammar, where $\mathcal{S} = \{+1, -1\}$. A dyck sequence must have even number of tokens $s_1, \ldots, s_{2k}$, and satisfy $\sum_{j=1}^{i} s_j \geq 0$ for $i \in [2k]$. Also it requires $\sum_{j=1}^{2k} s_i = 0$. This can be regarded as a valid operation sequence for a stack where $+1$ means push and $-1$ means pop. The requirement essentially means the stack cannot pop if it is empty, also it needs to be empty at start and end. The rendering function is similar as parity. In our experiment we use left and right parenthesis to represent $+1$ and $-1$, respectively.

As for denoising networks, we use attention-augmented UNet, where each block is equipped with linear attention and middle bottleneck equipped with full attention. The image size is 64 and the initial hidden width is 64, which means the bottleneck dimension is 512. We also adopted standard DiT-B model with hidden size 384 and patch size 8. We train with Adam optimizer with $lr = 8 \times 10^{-5}$, batch size $bs = 16$, total schedule ranging from 160k to 700k iterations.

**Training Schedule and Model Details** We use mainly two types of model for training in our experiments, namely DiT and UNet augmented with attention. The DiT is standard DiT-S model, with $33M$ parameter. The UNet initial channel is 64, and total parameter is $\sim 35.7M$. We training the score matching objective with equal $\lambda_t$. The training batch size is 16, 180k iteration for Quater-MNIST and $1.1M$ iteration for parity parathensis images.

### A.4 THE RESULTS ON RECENT MODELS

We also conduct experiment on most recent models such as StableDiffusion 3.5-medium (Esser et al., 2024) and FLUX1-dev. We use prompt that requires the model to generate rich text content without

specifying concrete words. For instance, "A piece of calligraphy art", "A newspaper reporting news", "A blackboard with formulas". And here are the test results, we can see that text hallucination is still ubiquitous. The seeds of six images of each model under same prompt are from 0 to 2, so all people can reproduce these results.

We also tried prompt to specify the content, i.e. "A paper saying 'To be or not to be, it is a question'." The results are plotted below. We can see both SD3.5 and FLUX1 still have text issues. SD3.5 has missing words or incorrect spelling more often.

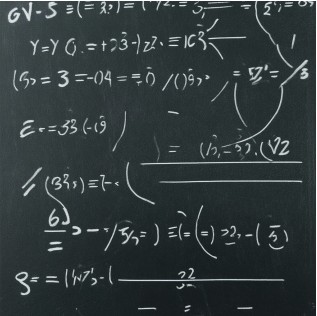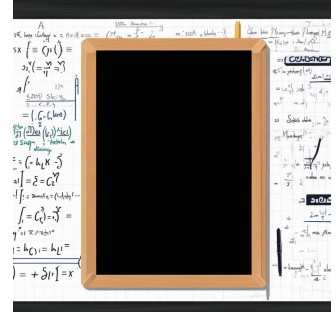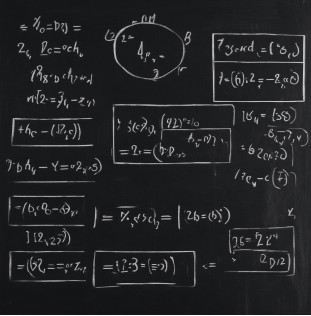

Figure 8: Visualizations of StableDiffusion 3.5's results on prompt "A blackboard with formulas"

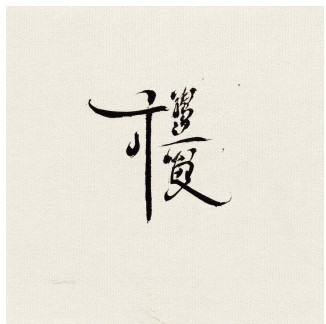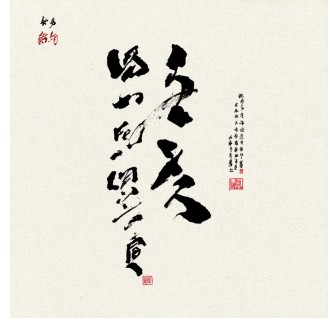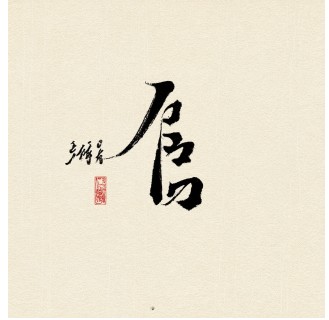

Figure 9: Visualizations of StableDiffusion 3.5's results on prompt "A piece of calligraphy art."

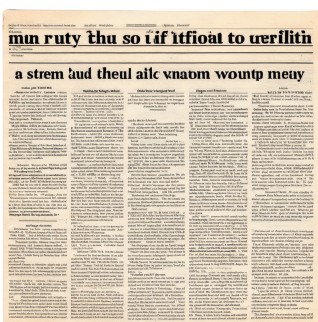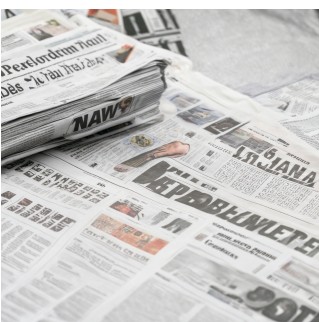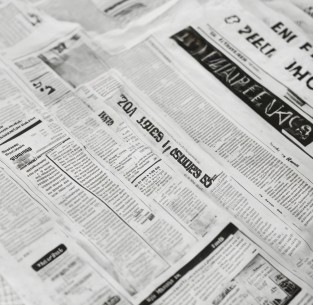

Figure 10: Visualizations of StableDiffusion 3.5's results on prompt "A newspaper reporting news."

## A.5 VISUALIZATIONS

In this part, we will show detailed visualizations of our experiments. For each experiment, we visualize following

- Generated hallucination samples.
- The trend of LDR along training process.
- The heatmap of LDR at some critical denoising timestep.

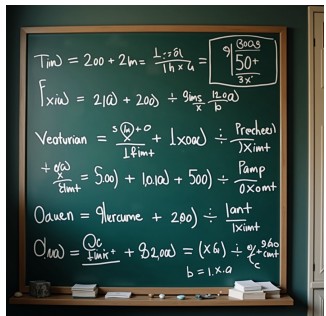 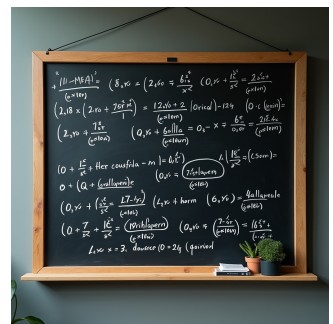 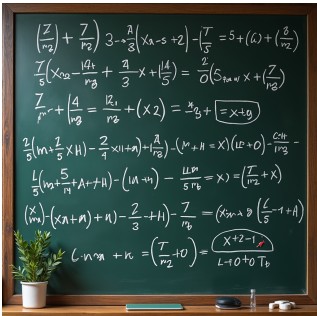

Figure 11: Visualizations of FLUX1's results on prompt "A blackboard with formulas"

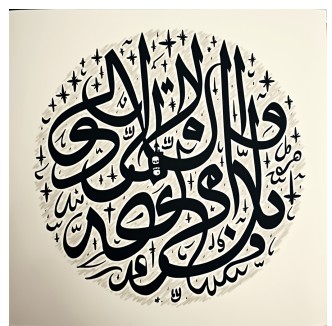 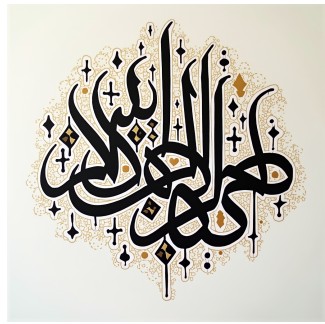 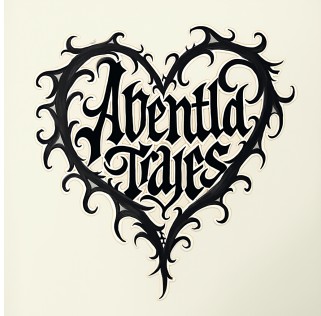

Figure 12: Visualizations of FLUX1's results on prompt "A piece of calligraphy art."

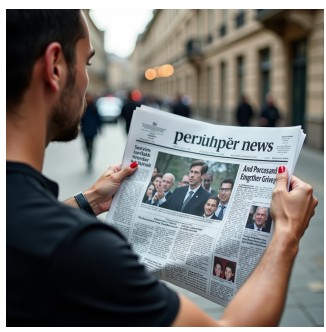 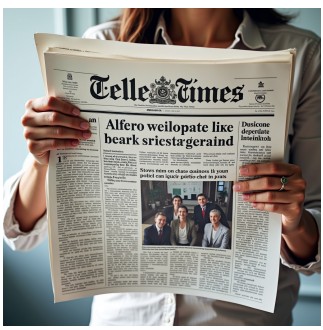 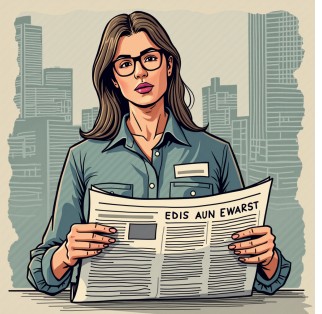

Figure 13: Visualizations of FLUX1's results on prompt "A newspaper reporting news."

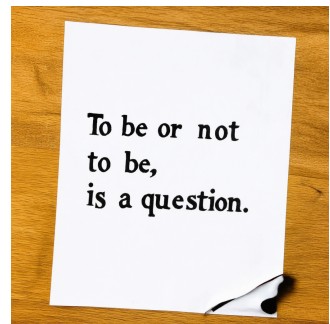

Figure 14: Visualizations of StableDiffusion 3.5's results on prompt "A paper saying 'To be or not to be, it is a question'."

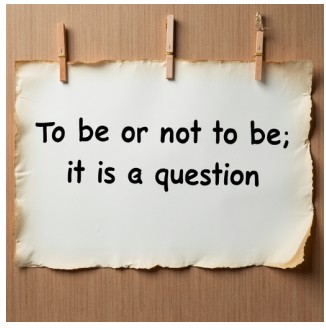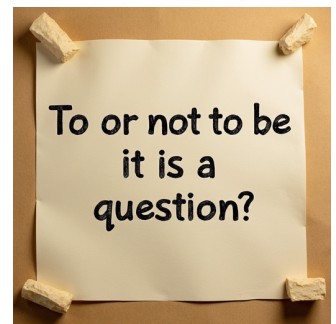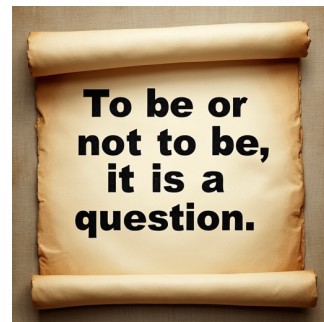

Figure 15: Visualizations of FLUX1's results on prompt "A paper saying 'To be or not to be, it is a question'."

## A.6 LDR ANALYSIS FOR FLUX-1-DEV AND STABLEDIFFUSION 3.5

we also conduct proposed LDR analysis on current models such as SD3.5 and FLUX1. However, Exact calculation the Jacobian matrix $J_{R,\theta}(x)$ requires large memory consumption for these enormous models with billions of parameters. We use zeroth order approximation to test LDR.

Note that $\mathbb{E}_\epsilon[\|f(x_0 + \epsilon) - f(x_0)\|^2] \approx \mathbb{E}_\epsilon \langle \epsilon, \nabla_{x_0} f(x) \rangle^2$. Therefore, we can set $\epsilon_1 \sim \mathcal{N}(0, \epsilon I_d)$ and get $\mathbb{E}_{\epsilon_1} \langle \epsilon_1, \nabla_{x_0} f(x) \rangle^2 = \epsilon^2 d \|\nabla_{x_0} f(x)\|^2$. Then set $\epsilon_2 \sim \mathcal{N}(0, \epsilon P_R P_R^\top)$ to get $\mathbb{E}_{\epsilon_2} \langle \epsilon_2, \nabla_{x_0} f(x) \rangle^2 = \epsilon^2 d_R \|P_R \nabla_{x_0} f(x)\|^2$. Taking ratio of these two terms and multiplying $d/d_R$, we obtain an approximation of LDR. We test the average LDR with prompting "A blackboard with formulas". Total reverse timesteps are 50. Here are the approximated LDR results:

| Timestep | 50 | 40 | 30 | 20 | 10 |
|---|---|---|---|---|---|
| **FLUX1** | 0.9253 | 0.8078 | 0.7992 | 0.8208 | 0.7994 |
| **SD3.5** | 0.9692 | 0.8283 | 0.7912 | 0.5726 | 0.4534 |

Table 1: LDR Results for FLUX1 and SD3.5

Apart from SD3.5 has a small LDR near the end (we do not know why), most LDR is relatively high, which corroborates our finding. This means when generating text content images, the diffusion model tend to care only local region for each part, without caring about underlying relation.

### A.6.1 PARITY PARENTHESIS

Please refer to figure 7 for details of generated hallucination sample and LDR analysis. The LDR heatmap is in figure 8.

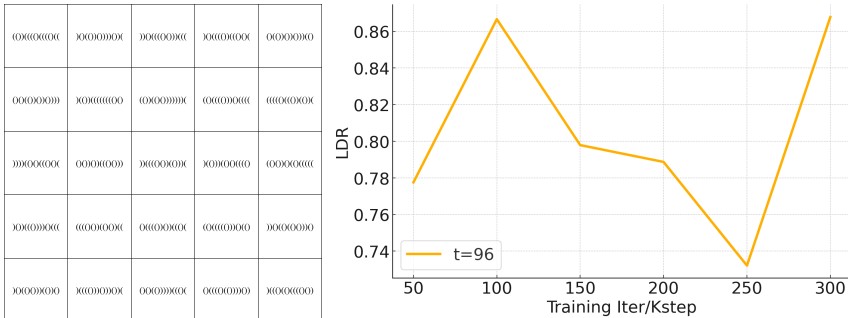

Figure 16: Some examples of generated hallucination samples at 300k steps. Note that only half of them satisfy parity constraint (even number of both parenthesis). The LDR at $\sqrt{\bar{\alpha}_t} = 0.1$ is high through all training procedure.

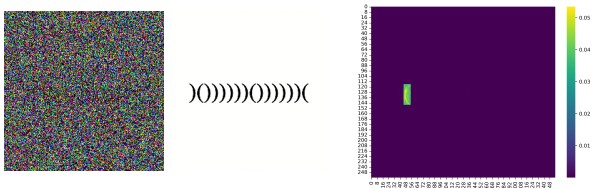

Figure 17: An example of $x_t, x_0$ and LDR heatmap. The LDR in this image is 0.9736 and the reference region is the second parenthesis. We can see the denoising model primarily only focues on this parenthesis' region to generate it. Therefore all the symbols are generated independent and fail to satisfy parity constraint.

### A.6.2   DYCK PARENTHESIS.

Perhaps surprisingly, we found that UNet model is capable of generating valid dyck sequences. After 60k iterations, the UNet model drops down and the accuracy for generated image increases. This can also be validated from probing a parenthesis' region to see which part of input noise the model is looking at. We found that model will only focus on local noise at hallucination phase, resulting in a high LDR. And when it overfits the data, the saliency map spreads globally and LDR decreases. See figure 10 for more details.

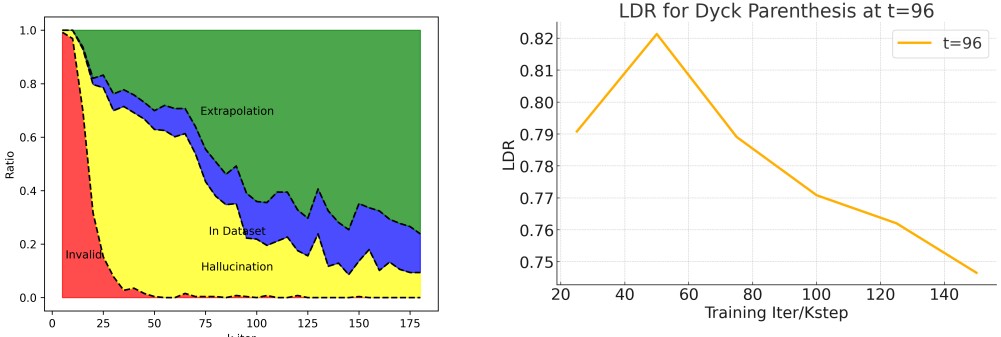

Figure 18: Generation proportion graph and LDR for $t = 96$. The reference region is the position at second parenthesis. Although seemed difficult, Dyck grammar is actually much easier to learn and extrapolate, since there are strong correlations between parenthesis. Interesting, there is still a hallucination phase, and hallucinations fades as LDR decreases.

### A.6.3   QUARTER MNIST.

The LDR analysis is shown in main content. Here we show some hallucinated generation and heatmap of LDR. As shown in figure 11, we can see that both UNet and DiT generate the top-left digit solely by local region's noise. As a consequence, these four digits are generated independently, therefore can not capture the innate relationship and rules.

### A.6.4   ENGLISH WORD AND CHINESE CHARACTERS.

Does hallucination in real-world text distribution also stem from local generation bias? We run experiments to verify this mechanism with image distribution contains common English words and Chinese characters. These English words are

[a, abandon, ability, able, about, above, accept, according, account, across, act, action,activity, actually, add, address, administration, admit, adult, affect, after, again,against, age, agency, agent, ago, agree, agreement, ahead, air, all, allow, almost, alone,along, already, also, although, always, American, among, amount, analysis, and, animal, another, answer, any, anyone, anything, appear, apply, approach, area, argue, arm, around,arrive, art, article, artist, as, ask, assume, at, attack, attention, attorney, audience,author, authority, available, avoid, away, baby, back, bad, bag, ball, bank, bar, base,

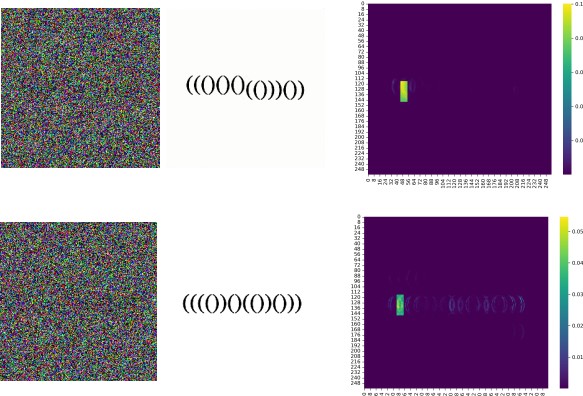

Figure 19: The LDR analysis for 20k training steps (first row, in hallucination) and 170k training steps (second row, correctly extrapolate). We can see a discrepancy for model's behaviors in terms of local v.s. global dependency. When model learns to correctly generate symbols, it will attend to overall region for coordinating different symbols, which means LDR is low. From left to right columns are $x_t, x_0$ and LDR heatmap.

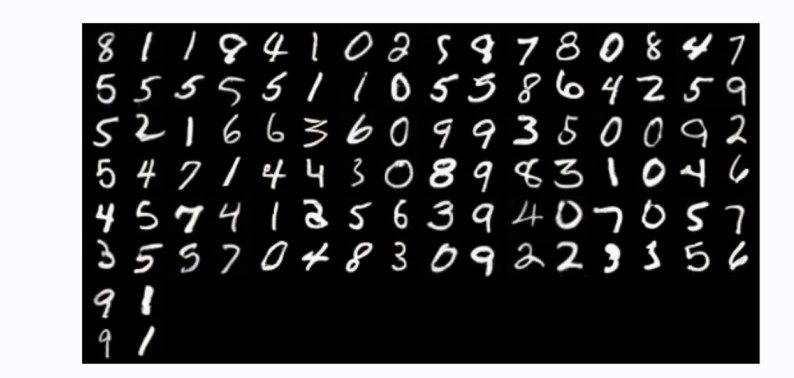

Figure 20: DiT generated Hallucinated samples for Quarter-MNIST dataset. Each four digit form a sample

be,beat,beautiful, because, become, bed, before, begin, behavior, behind, believe, benefit, best,better, between, beyond, big, bill, billion, bit, black, blood, blue, board, body,book,born, both, box, boy, break, bring, brother, budget, build, building, business, but,buy,by, call, camera, campaign, can, cancer, candidate, capital, car, card, care, career,carry,case, catch, cause, cell, center, central, century, certain, certainly, chair, challenge,chance, change, character, charge, check, child, choice, choose, church, citizen, city,civil,claim, class, clear, clearly, close, coach, cold, collection, college, color, come,commercial,common, community, company, compare, computer, concern, condition, conference, Congress,consider,consumer, contain, continue, control, cost, could, country, couple, course, court, cover,create, crime, cultural, culture, cup, current, customer, cut, dark, data, daughter,day,dead, deal, death, debate, decade, decide, decision, deep, defense, degree, Democrat,democratic,describe, design, despite, detail, determine, develop, development, die, difference,different,difficult, dinner, direction, director, discover, discuss, discussion, disease, do, doctor,dog, door, down, draw, dream, drive, drop, drug, during, each, early, east, easy,eat,economic, economy, edge, education, effect, effort, eight, either, election, else,employee,end, energy, enjoy, enough, enter, entire, environment, environmental, especially,establish,even, evening, event, ever, every, everybody, everyone, everything, evidence, exactly,example,executive, exist, expect, experience, expert, explain, eye, face, fact, factor, fail,fall,family, far, fast, father, fear, federal, feel, feeling, few, field, fight, figure,fill,film, final, finally, financial, find, fine, finger, finish, fire, firm, first, fish,five,floor, fly, focus, follow, food, foot, for, force, foreign, forget, form, former,forward,four, free, friend, from, front, full, fund, future, game, garden, gas, general,generation,get, girl, give, glass, go, goal, good, government, great, green, ground,

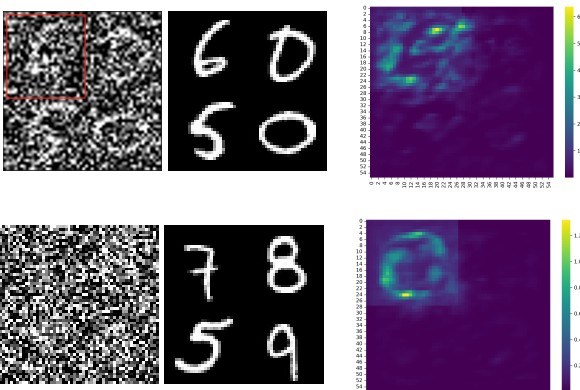

Figure 21: The LDR analysis for UNet (top row) and DiT (second) learning Quarter-MNIST dataset. $\sqrt{\bar{\alpha}_t} = 0.1$ and the reference region is top-left quarter. From left to right columns are $x_t, x_0$ and LDR heatmap.

group, grow,growth, guess, gun, guy, hair, half, hand, hang, happen, happy, hard, have, he,head,health, hear, heart, heat, heavy, help, her, here, herself, high, him, himself, his,history,hit, hold, home, hope, hospital, hot, hotel, hour, house, how, however, huge, human,hundred,husband, I, idea, identify, if, image, imagine, impact, important, improve, in, include,including, increase, indeed, indicate, individual, industry, information, inside, instead,institution,interest, interesting, international, interview, into, investment, involve, issue, it, item,its,itself, job, join, just, keep, key, kid, kill, kind, kitchen, know, knowledge, land,language,large, last, late, later, laugh, law, lawyer, lay, lead, leader, learn, least, leave,left,leg, legal, less, let, letter, level, lie, life, light, like, likely, line, list,listen,little, live, local, long, look, lose, loss, lot, love, low, machine, magazine, main,maintain,major, majority, make, man, manage, management, manager, many, market, marriage, material,matter,may, maybe, me, mean, measure, media, medical, meet, meeting, member, memory, mention,message,method, middle, might, military, million, mind, minute, miss, mission, model, modern,moment,money, month, more, morning, most, mother, mouth, move, movement, movie, Mr, Mrs,much, music,must, my, myself, name, nation, national, natural, nature, near, nearly, necessary,need, network,never, new, news, newspaper, next, nice, night, no, none, nor, north, not, note,nothing, notice,now, n't, number, occur, of, off, offer, office, officer, official, often, oh, oil,ok, old,on, once, one, only, onto, open, operation, opportunity, option, or, order,organization, other,others, our, out, outside, over, own, owner, page, pain, painting, paper, parent,part, participant,particular, particularly, partner, party, pass, past, patient, pattern, pay, peace,people, per,perform, performance, perhaps, period, person, personal, phone, physical, pick, picture,piece, place,plan, plant, play, player, PM, point, police, policy, political, politics, poor,popular, population,position, positive, possible, power, practice, prepare, present, president, pressure,pretty, prevent,price, private, probably, problem, process, produce, product, production, professional,professor, program,project, property, protect, prove, provide, public, pull, purpose, push, put, quality,question, quickly,quite, race, radio, raise, range, rate, rather, reach, read, ready, real, reality,realize, really,reason, receive, recent, recently, recognize, record, red, reduce, reflect, region,relate, relationship,religious, remain, remember, remove, report, represent, Republican, require, research,resource, respond,response, responsibility, rest, result, return, reveal, rich, right, rise, risk, road,rock, role,room, rule, run, safe, same, save, say, scene, school, science, scientist, score, sea,season, seat,second, section, security, see, seek, seem, sell, send, senior, sense, series, serious,serve, service,set, seven, several, sex, sexual, shake, share, she, shoot, short, shot, should,shoulder, show, side,sign, significant, similar, simple, simply, since, sing, single, sister, sit, site,situation, six, size,skill, skin, small, smile, so, social, society, soldier, some, somebody, someone,something, sometimes,son, song, soon, sort, sound, source, south, southern, space, speak, special, specific,speech, spend,sport, spring, staff, stage, stand, standard, star, start, state, statement, station,stay, step, still,stock, stop, store, story, strategy, street, strong, structure, student, study, stuff,style, subject, success,successful, such, suddenly, suffer, suggest, summer, support, sure, surface, system,table, take, talk, task,tax, teach, teacher, team, technology, television, tell, ten, tend, term, test, than,thank, that, the, their,them, themselves, then, theory, there, these, they, thing, think, third, this, those,though, thought, thousand,threat, three, through, throughout, throw, thus, time, to, today,

together, tonight,too, top, total, tough, toward,town, trade, traditional, training, travel, treat, treatment, tree, trial, trip, trouble,true, truth, try, turn,TV, two, type, under, understand, unit, until, upon, use, usually, value, various,very, victim, view, violence,visit, voice, vote, wait, walk, wall, want, war, watch, water, way, we, weapon,wear, week, weight, well, west,western, what, whatever, when, where, whether, which, while, white, who, whole, whom,whose, why, wide, wife, will,win, wind, window, wish, with, within, without, woman, wonder, word, work, worker,world, worry, would, write,writer, wrong, yard, yeah, year, yes, yet, you, young, your, yourself].

| shous | stan | ceraliyee | sonk | deart |
|---|---|---|---|---|
| no | scicon | cale | conpient | conglounment |
| sonde | stepe | exnage | tchetorn | riit |
| sentor | mor | caour | hold | mong |
| suriese | camer | trore | nappers | dffecn |

| profesing | methor | simder | pubility | Amerenal |
|---|---|---|---|---|
| seven | laok | frat | nex | gmen |
| action | hictum | yert | ndee | atothen |
| group | thme | freal | must | paer |
| nem | such | baite | fous | prwyer |

Figure 22: Diffusion generated results when trained on English common words' image (first row) and Chinese characters (second row). We find similar misspelling phenomenon for English generation and glyph by randomly assembling radicals in Chinese characters.

Also we construct a dataset using 3,000 common Chinese characters and render them in Kai font images. We use UNet to learn to generate images of these texts. The early stage generation results are shown in figure 14. Interestingly, we observe very similar pattern as in modern large scale diffusion models like StableDiffusion and Midjourney in our synthetic experiment. We probe denoising model at stage when it has hallucination, and finds that they all have very high LDR, indicating they generate letter or radicals independently and combine them.

## A.7   VALIDATION OF THEORETICAL FINDINGS

In this section, we corroborate our theoretical findings by experiments. We set $d = 8, 16$ and learn just the first dimension of score function for parity points using a two-layer ReLU-activated MLP. The model has 2000 hidden neurons and we set initialization scheme $\sigma_{init} = 1e - 3, b_i(0) = 0, a_i = \|w_i\|^2$. We train with small learning rate $\eta = 1e - 6$ and discover following interesting phenomena.

- The loss curves exhibit a stair-like shape, meaning it has three phases.

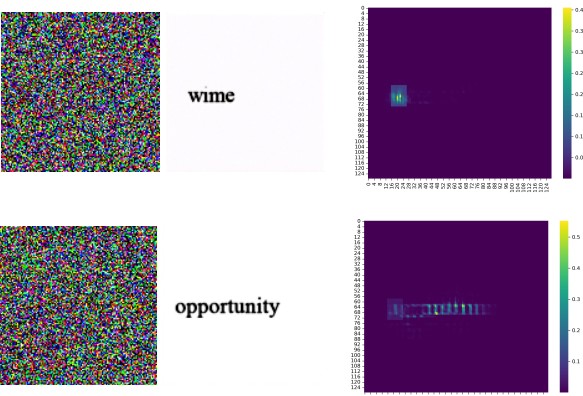

Figure 23: The LDR analysis for model at 20k steps (first row) and 200k steps (second row) learning on common English words dataset. $\sqrt{\bar{\alpha}_t} = 0.1$ and the reference region is the first and second letter. We can see that when model hallucinates, it only attends to local region, therefore randomly spelling the letters. It will account globally when overfitting to reproduce words within the training dataset. From left to right columns are $x_t, x_0$ and LDR heatmap.

- These three phases correspond to best linear interpolation, best univariate interpolation, and optimal approximator.
- At initial stage, the network's weight $w_i$ aligns well with $e_1$, and stick with this state through the first and second stage.

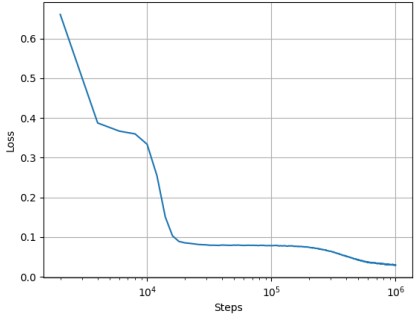

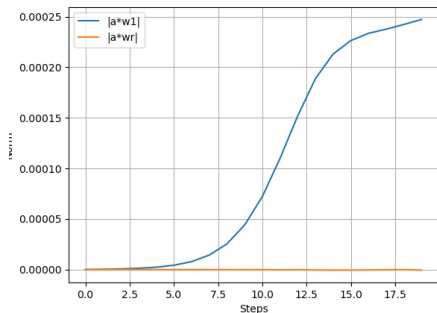

(a) Three stairs shape loss. Each stage represents a saddle point.

(b) The average norm for the first dimension and the rest of weight parameter among hidden neurons. In the first stage, the model only extracts the input's first dimension's information, resulting in a local and sparse input dependency.

As shown in figure 17. While the ground truth score function is not a univariate function of $x^{(1)}$ as in left. The shaded area near the origin means the score function has also dependency on other input dimensions $x^{(j)}, j > 1$. However,The MLP is biased towards learning a univariate function. Even though MLP has access to value from all input dimensions. This results in a local generation bias and let the model independently sample each dimension. Therefore this model essentially samples on all vertices on hypercube rather than parity points.

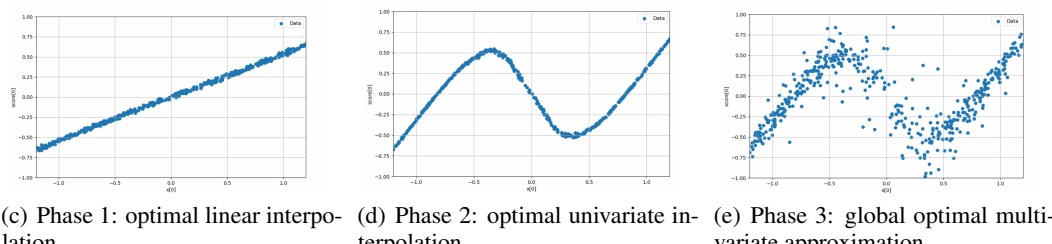

(c) Phase 1: optimal linear interpolation.

(d) Phase 2: optimal univariate interpolation.

(e) Phase 3: global optimal multivariate approximation.

Figure 24: Three-phase functionality of learned MLP score network. The x-axis is $x^{(1)}$. The MLP performs local generation, if its output against $x^{(i)}$ is nearly a function curve with no ambiguity in mapping.

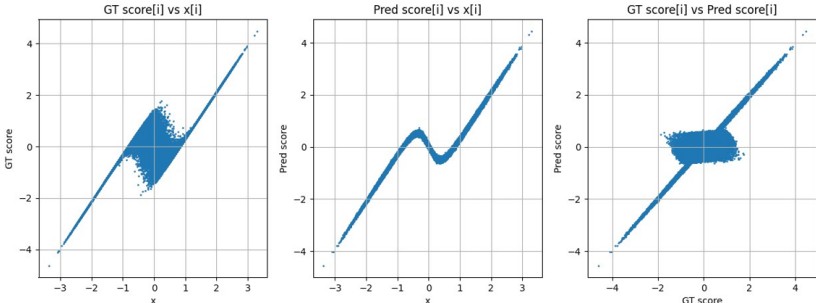

Figure 25: The local generation bias in MLP.

