# OpenReview forum: "Towards Understanding Text Hallucination of Diffusion Models via Local Generation Bias"
_ICLR.cc/2025/Conference — ICLR 2025 Poster_

### Official Review · Reviewer_MEJz · 2024-10-31

**Soundness:** 4
**Presentation:** 2
**Contribution:** 3
**Rating:** 6
**Confidence:** 3

**Summary:**

This paper investigates textual hallucinations, where diffusion models generate locally accurate but globally nonsensical symbols. The authors introduce the local dependency ratio (LDR) as a probe to quantitatively measure the model's dependency on local regions when conducting denoising. With LDR, the authors observe that the textual hallucinations are caused by the local generation bias, And this phenomenon exists across various denoising network architectures including MLP and transformers. The authors also provide a theoretical analysis of its underlying mechanism.

**Strengths:**

1. The authors explore textual hallucination in generative models, addressing a crucial yet often overlooked research problem.

2. They introduce the local dependency ratio (LDR) to examine how diffusion models depend on localized regions of input noise during denoising and generation. The analysis is both robust and engaging.

3. They present a sound theoretical examination of the mechanism underlying textual hallucination.

**Weaknesses:**

1. Labeling this issue as "textual hallucination" might cause confusion with hallucination problems in large language models (LLMs). It’s advisable to adjust the task description or add a notice at the start of the main paper for clarity.

2. Following up on the previous question, would this issue also occur in Vision Language Models (MLLMs)?

3. The experimental setup appears less practical and somewhat limited in scope. Most experiments were conducted on small, toy datasets and within narrow domains, such as digit generation. Could larger datasets and real-world applications, like image generation on ImageNet, be explored? Additionally, would textual hallucination issues appear in other real-world contexts?

4. The theoretical analysis is constrained to two-layer MLP learning parity points on a hypercube, creating a significant gap between MLPs and more complex models like UNet and DiT. The relevance of the proposed theory to real-world models is therefore uncertain and requires further investigation.

5 It is suggested to include potential strategies to address textual hallucination issues. Although fully solving this problem may be challenging, proposing ways to partially mitigate this limitation in generative diffusion models would be valuable.

**Questions:**

Please refer to the weaknesses section.

---

> ### Author Response · Authors · 2024-11-20
> **Response to Reviewer MEJz**
>
> Thank your time and thoughtful suggestions! Here are the responses to your problems.
>
>
> **1. Limited Experiments.**
> > The experimental setup appears less practical and somewhat limited in scope. Most experiments were conducted on small, toy datasets and within narrow domains, such as digit generation. Could larger datasets and real-world applications, like image generation on ImageNet, be explored? Additionally, would textual hallucination issues appear in other real-world contexts?
>
> The reason we use synthetical dataset is to gain a well-controlled data setting for clean and comprehensive study, therefore the lack of practical meaning is inevitable. Nevertheless, we *do* verify our findings and insights in more practical scenarios in appendix. For instance, we rendered 1000 common English words and Chinese characters and conduct similar study, discovering that a high LDR is also spotted. And the local generation bias persists.
>
> We also conduct LDR analysis for current models such as FLUX1 and SD3.5, please refer to general response and appendix.
>
> **2.Does VLM have similar issues?**
> > Labeling this issue as "textual hallucination" might cause confusion with hallucination problems in large language models (LLMs). It’s advisable to adjust the task description or add a notice at the start of the main paper for clarity.
> Following up on the previous question, would this issue also occur in Vision Language Models (MLLMs)?
>
> It is interesting and meaningful to investigate whether VLM also exhibits such issues. However, as VLM is not generative model, we can not directly sample from its internal model to visualize its understanding on texts. But there does exist concurrent work[1] finding that VLM is not so good at recognizing text content, i.e. OCR task.
>
> [1] Vision language models are blind. *Pooyan Rahmanzadehgervi, Logan Bolton, Mohammad Reza Taesiri, Anh Totti Nguyen*
>
> **3. Limitation of Theoretical Study.**
>
> Thank you for your suggestion. Please refer to general response 4.

---

> ### Author Response · Authors · 2024-11-24
> **Kind Reminder: Follow-up on Rebuttal Feedback**
>
> Thank you for your thoughtful and constructive feedback on our submission. We have carefully addressed your concerns and provided detailed responses in our rebuttal. We kindly remind you to review our rebuttal at your earliest convenience. If there are any remaining issues or further clarifications needed, we would be more than happy to address them.

---

> ### Author Response · Authors · 2024-12-01
>
> Thank you very much for your positive rate and valueable feedback. In our rebuttal, we have further addressed your concerns with detailed explanation and results. If you have any further questions, we are more than glad to answer them.
>
> Also, we would sincerely appreciate it if you could increase your score. A clearer acceptence would make AC's job much easier for current boarderline score and enable this work get opportunity to be exposed at conference. Thank you once again!

---

### Official Review · Reviewer_KNEH · 2024-11-01

**Soundness:** 3
**Presentation:** 3
**Contribution:** 3
**Rating:** 6
**Confidence:** 4

**Summary:**

This paper explores the score-based diffusion models' text hallucination, where each generated symbol is accurate but arranged nonsensically. Through the experiments on various settings of toy datasets, the authors observed that this phenomenon is related to the local generation bias of the score model, which is the tendency of the score model to not regard the entire input region and their correlation.
In the process, a metric called Local Dependency Ratio(LDR) was newly suggested to quantify how the diffusion model relies on specific local region of the input for denoising.

The authors also found that this bias emerges across various architectures, including CNN, MLP, and transformers.
Lastly, they examined training dynamics on specifically designed data to propose a theoretical perspective on the emergence of local generation bias.

**Strengths:**

-The paper newly identifies the possible cause of the text hallucination, with extensive experiments over various dataset settings of symbols and their correlation.

-The suggested metric of LDR is intuitive and shows good explainability of the score model behavior.

**Weaknesses:**

-Since most of the results are obtained from the simple controlled data, there's still some gap to fill to extend the results into the real-world language data.

-The proposed theoretical analysis on local generation bias with training dynamic is limited for the used problem setting(e.g. most conclusions in Section 5.1 require Assumption 5.1).

**Questions:**

-In the dyck parenthesis experiment in Section A.4.2, it seems like the diffusion model correctly figures out the grammar beneath valid dyck sequences.  Does this suggest the denoising network can overcome the local generation bias and avoid hallucination, contrary to the paper's conclusion? If not, what is the difference between this experiment and other results?

-In Figure 5(right), how does the figure contain the LDR plot with t=121 and t=168 if the total timestep is 100?

-It would be better if there's an LDR comparison on the sampling of pre-trained models, between generating hallucinated samples and generating correct samples.

-In the overall results, the authors distinguished the 'extrapolated' samples and 'in-dataset memorized' samples. However, in terms of resolving hallucinations, is there a meaningful difference between the generation of extrapolated data and memorized data? Memorized output is also valid, and memorization by overfitting may not be due to the intrinsic symbol correlation itself, but to other factors such as training data preparation, model size, etc. Also, for some datasets, especially real-world texts, the correct extrapolation from the training dataset is almost impossible, and being contained in the database becomes the only way to be valid.

---

> ### Author Response · Authors · 2024-11-20
> **Response to Reviewer KNEH**
>
> Thank your time and thoughtful suggestions! Here are the responses to your problems.
>
> **1. Difference between Dyck and Parity.**
>
> >In the dyck parenthesis experiment in Section A.4.2, it seems like the diffusion model correctly figures out the grammar beneath valid dyck sequences. Does this suggest the denoising network can overcome the local generation bias and avoid hallucination, contrary to the paper's conclusion? If not, what is the difference between this experiment and other results?
>
> That is a good question. First, Dyck does not satisfy one of our central assumption-**the pairwise independence of tokens**. Any two tokens in parity distribution have independent marginal distribution. In real-world text, different letters of a text (especially long range) are also nearly independent. But Dyck grammar has strong correlation between context and each token. For example, if there have already been a lot of ( in the prefix, then the chance of having a ) would be much higher than (.
> Moreover, even the model can successfully generate Dyck sequences, it still can not uniformly cover all the unseen data to correctly extrapolate. Therefore, the success of Dyck is subtle, model learns some relation and is able to combine valid sub-sequences, but it still does not fully understand it.
>
> **2. Explanation on Figure 5 and Experiment on Real Models.**
> > It would be better if there's an LDR comparison on the sampling of pre-trained models, between generating hallucinated samples and generating correct samples.
>
> That is a very good point. We renew figure 5 and put explanation in general response 3. Please refer to it.
>
> **3. Why Distinguish In-Dataset and Extrapolation?**
> > in terms of resolving hallucinations, is there a meaningful difference between the generation of extrapolated data and memorized data? Memorized output is also valid, and memorization by overfitting may not be due to the intrinsic symbol correlation itself...Also, for some datasets, especially real-world texts, the correct extrapolation from the training dataset is almost impossible, and being contained in the database becomes the only way to be valid.
>
> We acknowledge that unconditional extrapolation on complex real-world distributions, such as text, may not be meaningful for all the generation tasks. However, this is beyond the scope of the current paper.   The main point of the paper is that the models we concern do not have the flexibility to enable extrapolated generation.  This implies that  the models cannot really *understand* the underlying grammar, and we need to use them with caution in practice.
>
> Moreover, The core objective of developing generative models is to create content that extends beyond the specific instances present in the training dataset, otherwise we just need to retrieve nearest neighbor from the dataset. Although focusing on text issues, the discovered local generation bias may have broarder and deeper meaning in other artifacts. All these requires a clear distinguishment between in-dataset sample and extrapolated ones.

---

> ### Author Response · Authors · 2024-11-24
> **Kind Reminder: Follow-up on Rebuttal Feedback**
>
> Thank you for your thoughtful and constructive feedback on our submission. We have carefully addressed your concerns and provided detailed responses in our rebuttal. We kindly remind you to review our rebuttal at your earliest convenience. If there are any remaining issues or further clarifications needed, we would be more than happy to address them.

---

> > ### Comment · Reviewer_KNEH · 2024-11-25
> > **Reply to the authors' response**
> >
> > I thank the authors for their revision, and I believe the authors have resolved my questions and concerns.
> >
> > I still suggest that the manuscript be above the acceptance threshold, and keep my rating of 6.

---

> > > ### Author Response · Authors · 2024-12-01
> > >
> > > Thank you very much for your positive rate and valueable feedback.Your constructive advice on empirical validation and correcting minor details help us improve our draft greatly.
> > >
> > > Also, we would sincerely appreciate it if you could increase your score. A clearer acceptence would make AC's job much easier for current boarderline score and enable this work get opportunity to be exposed at conference. Thank you once again!

---

### Official Review · Reviewer_obGg · 2024-11-03

**Soundness:** 3
**Presentation:** 2
**Contribution:** 3
**Rating:** 6
**Confidence:** 4

**Summary:**

The paper performs a structured and theoretically motivated investigation of hallucination in text generation with diffusion models.
The authors construct two toy problems, parity parenthese and quarter MNIST, to investigate the phenomenon. The empirical evaluation along training steps and different generative architectures reveals that DMs fail to learn complex rules governing symbol distributions without access to strong priors. Using the proposed Local Dependency Ration (LDR) the authors demonstrate that symbols are generated with mainly localized attention, despite architecture design allowing for the capture of global structure.
The authors further underpin these empirical observations by providing theoretical proof that the DM training paradigm can get stuck at a local saddle set, which forces the model to learn marginal, localized distributions.

**Strengths:**

- **Key problem from a new perspective.** In investigating text generation with diffusion models, the authors work on a core problem in the field with high relevance. The more theoretical approach chosen by the paper offers a novel perspective and interesting insights.
- **Solid evaluation**. The authors construct two suitable proxy tasks for their analysis on which they perform compelling evaluations across training steps and architectures.
- **Novel probing tool.** The proposed LDR metric is an interesting approach that could potentially benefit other fields and aid future interpretability research

**Weaknesses:**

- **Generalization to current models**. While the paper does perform valuable evaluations and theoretical analysis on a constrained problem set, it fails to analyze strong, general-purpose models. Recent models like SD3, Flux, Playgroundv3, SD3.5, or Recraftv3 have become progressively better at text generation. At least if the text to be generated is supplied in the model prompt. Since many of these models' weights are openly accessible, it would be key for the paper's evaluation to investigate them using the proposed LDR metric. Specifically, insights into whether these models leverage more global structure or if improvements are tied to the strong priors of text conditioning
- **Readability.** In the same vein, the theoretical sections of the paper may be hard to follow for some readers, although the actual problems, like the proposed Quarter MNIST, are intuitive and easy to understand. The paper would benefit from interleaving some more plastic examples into the methodological definitions and theoretical analyses to appeal to a broader audience.
- **Lack of derived recommendations/future work.** The authors outline multiple issues in DMs that lead to hallucinations in text generation. Despite numerous insights, the paper does not come up with approaches or recommendations that would allow future work to address the same issues.

## Minor Comments
- The differing y-axis of the two plots in Fig. 5 is somewhat misleading.
- Similarly, the caption of Fig. 5 refers to a total $T=100$, although the right Figure goes up to $t=168$
- **Inconsistent Figure Placement.** The authors sometimes place Figures at the top of a page and, at other times, inline where the Figure is referred to in the main body. I'd recommend choosing one consistent placement.
- Typo in L493 "critical rule" instead of "critical role"


**Edit after rebuttal**

The authors rebuttal has addressed my main concerns. I have raised my score to a 6 according and vote for acceptance.

**Questions:**

- **Q1** When comparing UNet and DiT architectures, do the authors ensure equal parameter count? Otherwise, the difference in performance could be attributed to an increased capacity in favor of the DiT.
- **Q2** Could the authors further elaborate on the difference across timesteps in Fig. 5. First, could the authors clarify if in their notation the generative/reverse diffusion process goes from $T \rightarrow 0$. I.e. higher values of $t$ correspond to earlier/more noise steps in the denoising process. If that is the case, the results are somewhat counter-indicative to prior observations of diffusion models. It seems to be the current consensus that earlier steps generate larger/global image aspects with later steps refining localized features.
- **Q3** Following up on Q2, do the authors have an intuition/explanation why the UNet seems to have a clear progression over timesteps from between more global and local attention, whereas the DiT does not exhibit such a seperation?
- **Q4** Is the x-axis in Figure 5 the same as in Figure 4 as is implied in the main body? If so, why are the axes over the 2 Figures not aligned? It would make it easier for the reader to identify correlations in LDR with downstream performance.

---

> ### Author Response · Authors · 2024-11-20
> **Response to Reviewer obGg**
>
> Thank your time and thoughtful suggestions! Here are the responses to your problems.
>
> **1. Generalize Results to Current Models.**
> > ..., it would be key for the paper's evaluation to investigate them using the proposed LDR metric. Specifically, insights into whether these models leverage more global structure or if improvements are tied to the strong priors of text conditioning.
>
> We agree that testing our LDR metric on latest model is important to validate the intuition and proposed mechanism. We first sample images with rich text content generated by SD3.5 and FLUX1, and text issue is still happening. For more details, please refer to general response 2. We also conduct an approximated LDR analysis for recent models, for more details please refer to general response 3.
>
> The reason we use synthetical dataset is to gain a well-controlled data setting for clean and comprehensive study, therefore the lack of practical meaning is inevitable. Nevertheless, we *do* verify our findings and insights in more practical scenarios in appendix. For instance, we rendered 1000 common English words and Chinese characters and conduct similar study, discovering that a high LDR is also spotted. And the local generation bias persists.
>
> **2. Parameter and Overfitting.**
> > When comparing UNet and DiT architectures, do the authors ensure equal parameter count?
>
> Yes, please refer to general response 3. Actually, even successful memorization at large-scale model does not harm our argument. We do *not* mean that model can never learn global relation even by overfitting. But under relatively few data and insufficient training, hallucination persists to happen and we aim to understand its mechanism, namely local generation bias. Note that overfitting usually require 5x training iterations than just achieving low FID for plausible visual clues, where most real-world diffusion training stops here.
>
> **3. About Timestep and Locality.**
> > First, could the authors clarify if in their notation the generative/reverse diffusion process goes from T→0. I.e. higher values of t correspond to earlier/more noise steps in the denoising process. If that is the case, the results are somewhat counter-indicative to prior observations of diffusion models . It seems to be the current consensus that earlier steps generate larger/global image aspects with later steps refining localized features.
>
> We renew figure 5 and explain it in general response 3. The $t$ refers to forward procedure timestep, and denoising is considerd as $T\to 0$, higher $t$ means close to pure noise.
>
> On question about localized generation, you raised a very good question. Actually, the locality for DiT against timestep is a U-shape dependence. LDR is large for small and large timesteps $t$, but small for intermediate $t$. This is because for large $t$, the denoising function is close to identity $s_{\theta}(x)=x$, which is actually a localized function. Along the decrease of $t$, SNR increases and content gradually emerges, LDR decreases and model start to draw in sketch, caring more about global structure. Finally when $t$ is small, model only refine local details, thus LDR increases again.
>
> **4. Progression Difference.**
> > do the authors have an intuition/explanation why the UNet seems to have a clear progression over timesteps from between more global and local attention, whereas the DiT does not exhibit such a seperation?
>
> I guess you mean DiT has more clear regression in the local-to-global attention transition. This is because DiT is a pure-transformer structure operating on latent tokens, therefore its ability to overfit and memorize dataset (especially token's composition grammar) is stronger than attention-agumented UNet. But still, even equipped with this capacity, DiT struggled with local generation for a long time before it eventually memorize the dataset.
>
> **5. Typos and Typesetting.**
>
> Thank you for pointing out these problems! We renew figure 5 and correct typos in our draft. We also add more intuitive explanation for theory and will recommend more related work that potentially gives solution in our future versions.

---

> > ### Comment · Reviewer_obGg · 2024-11-25
> > **Thorough Response addresses main concerns**
> >
> > I thank the authors for their thorough response.
> >
> > Having read the comments and rebuttals to all other reviews and considering the revised version, I suggest accepting the revised paper to ICLR.
> >
> > I have adjusted my score accordingly.
> >
> >
> > P.S.
> > I was unable to find the empirical LDR results from general response 3 in the revised paper. I'd urge the authors to include these results along with a discussion on the observed U-Net shape of LDR over timesteps. These insights would be valuable to other readers of the paper as well.

---

> > > ### Author Response · Authors · 2024-11-25
> > > **Thank You For Your Suggestions**
> > >
> > > We sincerely appreciate your response. As for the empirical LDR results, we plan to elaborate a fine-grained authentic first-order gradient analysis with multiple GPUs, and add the results to our final public version. This takes some time.
> > > We appreciate your suggestion. These problems raised in discussion and observations will definitely be added in our final draft.
> > >
> > > Thank you again for your time and constructive feedback!

---

> ### Author Response · Authors · 2024-11-24
> **Kind Reminder: Follow-up on Rebuttal Feedback**
>
> Thank you for your thoughtful and constructive feedback on our submission. We have carefully addressed your concerns and provided detailed responses in our rebuttal. We kindly remind you to review our rebuttal at your earliest convenience. If there are any remaining issues or further clarifications needed, we would be more than happy to address them.

---

### Official Review · Reviewer_h9FB · 2024-11-03

**Soundness:** 3
**Presentation:** 3
**Contribution:** 2
**Rating:** 6
**Confidence:** 2

**Summary:**

This paper examines the issue of textual hallucinations in score-based diffusion models. The authors identify this phenomenon as stemming from a local generation bias, where the denoising networks rely heavily on highly correlated local regions of input data, leading to a failure in capturing the global structure. The paper proposes the Local Dependency Ratio (LDR) as a new metric to quantify this bias, demonstrating its persistence throughout training and across various network architectures. The findings suggest that the hallucinations are a result of the training dynamics rather than limitations in the model's architecture

**Strengths:**

1.This paper proposes a new metric for measuring text hallucinations.The introduction of Local Generation Bias and the Local Dependency Ratio (LDR) provides a new perspective on understanding hallucinations in diffusion models.
2.The paper features a rigorous derivation process, and its logic is sound and clearly structured.

**Weaknesses:**

1. Some of the core arguments are not well articulated:
The paper suggests that the causes of hallucinations stem from fundamental training dynamics rather than architectural limitations. However, the comparative experiments in Figure 4 only demonstrate that different model architectures exhibit hallucinations, it does not completely decouple architectural influences
2. Insufficient experiments: The article conducts comparative experiments under two different experimental settings on only two datasets. Including more validation experiments would enhance the reliability of the paper.

**Questions:**

1. I am trying to find some arguments to support the thesis in the paper that 'the causes of hallucinations stem from fundamental training dynamics rather than architectural limitations' , But I haven't found any direct relevant evidence. Could you please provide stronger evidence to support this point? Perhaps I miss some key information.
2. The experimental content included in the paper is still somewhat lacking. Could you provide experiments on more datasets (or more models) to demonstrate the generalizability of the method?

---

> ### Author Response · Authors · 2024-11-20
> **Response to Reviewer h9FB**
>
> Thank your time and thoughtful suggestions! Here are the responses to your problems.
>
> **1. About Architecture Influence.**
> > However, the comparative experiments in Figure 4 only demonstrate that different model architectures exhibit hallucinations, it does not completely decouple architectural influences.
>
> Yes, architecture does make a difference. It is impossible to rule out any potential model architecture which can achieve text generation, and the best we can do is to test this on all existing models to verify. For more information, please refer to general response 1 and 3.
>
> **2. Insufficient experiments.**
> > The article conducts comparative experiments under two different experimental settings on only two datasets. Including more validation experiments would enhance the reliability of the paper.
>
> In our appendix, we conduct experiments beyond Quater-MNIST and Parity Parenthesis, including Dyck, English words and Chinese characters. All of them yields similar conclusions. We also add new experiments on up-to-date models like SD3.5 and FLUX1 in appendix, and the local generation mechanism still holds. Please refer to general reponse 2.

---

> > ### Comment · Reviewer_h9FB · 2024-11-30
> > **Thanks for your rebuttal**
> >
> > Thank you very much for your response. Your answer has resolved my issue to some extent, and I will adjust my score accordingly.

---

> ### Author Response · Authors · 2024-11-24
> **Kind Reminder: Follow-up on Rebuttal Feedback**
>
> Thank you for your thoughtful and constructive feedback on our submission. We have carefully addressed your concerns and provided detailed responses in our rebuttal.
> We kindly remind you to review our rebuttal at your earliest convenience. If there are any remaining issues or further clarifications needed, we would be more than happy to address them.

---

### Official Review · Reviewer_yRki · 2024-11-04

**Soundness:** 2
**Presentation:** 2
**Contribution:** 3
**Rating:** 6
**Confidence:** 3

**Summary:**

This paper investigates the local generation bias phenomenon in diffusion models and its relationship to text hallucination. The authors propose the Local Dependency Ratio (LDR) metric to provide empirical insights into this bias and present theoretical analysis attempting to explain the underlying causes of hallucinations.

**Strengths:**

The paper offers a novel perspective on diffusion model hallucinations through the lens of local dependency analysis.
The introduction of the Local Dependency Ratio (LDR) metric, along with the Quarter MNIST dataset, provides a useful probe for evaluating models' ability to consider global information versus local information.

**Weaknesses:**

- The central claim about hallucinations stemming from training dynamics rather than architectural limitations lacks substantial support:
  - No proposed training paradigm to demonstrate improvement.
  - Recent work (Stable Diffusion 3 [1]) achieves significant typography improvements through architectural changes (MMDiT)
- Regarding the experiments in section 4.2, it is possible that the network capacity might be insufficient to learn global relations. There should be more details on the model architecture. (It is claimed that the details of the model architecture are in the appendix, but they don't.)
- In addition to the previous point, the DiT model seems to be able to memorize the training samples very well. The model does not need to learn the true latent relation ($s_1 + s_2 = s_3 + s_4$) if it can simply memorize all training samples. This mitigates the validity of the claim that diffusion models cannot learn the global relation.
- The description "both have even numbers" about the Parity Parenthesis dataset seems incorrect. The claim "random guess has 50% chance of satisfying parity requirement" seems questionable.
- The authors attempt to theoretically prove that neural networks are prone to learning the marginal distributions instead of the global correlations by analyzing a two-layer MLP. However, the analysis may not generalize to U-Net or Transformer-based diffusion models. The DiT model is able to memorize Quarter MNIST training samples, demonstrating its capability of learning more than the marginal distributions.

[1] Esser P, Kulal S, Blattmann A, et al. Scaling rectified flow transformers for high-resolution image synthesis[C]//Forty-first International Conference on Machine Learning. 2024.

**Questions:**

See weaknesses.

---

> ### Author Response · Authors · 2024-11-20
> **Response to Reviewer yRki**
>
> Thank your time and thoughtful suggestions! Here are the responses to your problems.
>
> **1. Insufficient Parameter to Learn Global Relation.**
> > Regarding the experiments in section 4.2, it is possible that the network capacity might be insufficient to learn global relations. There should be more details on the model architecture.
>
> The total number of parameters of attention augmented UNet of init_dim=64 (35.7 M) and DiT-S (33 M) are comparable. It should have been written in appendix and we apologize for our negligence. Model architecture does play a role, since DiT can eventually overfit but UNet can not. But both models undergo a hallucination phase and shares similar mechanism in terms of local generation, which is what we aim to understand. (Please refer to general response 3.)
>
> Also, according to our experiment, recent work (SD3.5, FLUX1) does not solve unconditional text generation task. Hallucination still exists, even for some conditional generation. Please refer to general response 2.
>
> **2. The Model Capture the Global Relation Eventually.**
> > The model does not need to learn the true latent relation (s1+s2=s3+s4) if it can simply memorize all training samples. This mitigates the validity of the claim that diffusion models cannot learn the global relation.
>
> Yes, we also confirm in our experiments that after elapsed time of training, the model can overfit and memorize the training data, which is impossible without capturing global relation. **But we are *not* claiming that model can never generate valid data even by memorizing, but to understand why model hallucinates at under-trained and insufficient-data (with respect to text generation) regime by generalizing incorrectly, which is real-world models like SD3.5 lie in.**
>
> **3. More Explanation on Parity Dataset.**
> > The description "both have even numbers" about the Parity Parenthesis dataset  seems incorrect. The claim "random guess has 50% chance of satisfying parity requirement" seems questionable
>
> In our experiments, there are total 8 or 16 number of tokens $x_i$. Therefore, parity condition $\prod_{i=1}^L x_i=1$ is equivalent to set $X^-=\\{i\in[L]:x_i=-1\\}$ has even number of elements. If that is the case, then $X^+ = \\{ i\in[L]:x_i=1 \\}$ will have $L-|X^-|$ number of elements, which is also even. Therefore, if all $x_i$ are randomly drawn from $\\{\pm 1\\}$, then with half chance $|X^-|$ is even, so independent random generation has half chance of being valid for parity.
>
> **4. Limitation of Theoretical Analysis.**
>
> Our theoretical study is a conceptual framework to provide insight rather than rigorous disproof. Please refer to general response 4.

---

> ### Author Response · Authors · 2024-11-24
> **Kind Reminder: Follow-up on Rebuttal Feedback**
>
> Thank you for your thoughtful and constructive feedback on our submission. We have carefully addressed your concerns and provided detailed responses in our rebuttal.
> We kindly remind you to review our rebuttal at your earliest convenience. If there are any remaining issues or further clarifications needed, we would be more than happy to address them.

---

> > ### Comment · Reviewer_yRki · 2024-11-27
> >
> > Thank you for the responses. I agree that this paper provides intriguing insights into potential causes of the hallucination phenomenon. However, I still hold the opinion that better experiments could have been designed and conducted. More importantly, the authors have not addressed my concern about the overly strong central claim that hallucinations stem from training dynamics rather than architectural limitations. This might be addressed by proposing a new training paradigm to mitigate hallucination. Otherwise, I would suggest to tone down the central claim.

---

> ### Author Response · Authors · 2024-11-27
>
> Thank you for your further response. Actually we did have a response on supporting our claim in general response 1 to 4. To summarize, we further foster our claim from following perspectives
>
> * **Empirical Evaluation** In the rebuttal (general response 2,3), we add new test of empirical large model like SD3.5 and FLUX1 to the original paper, finding that their unconditional text generation still suffers from hallucination issue. (appendix, page 18-20) We also conduct an approximated version of our LDR analysis for these modern models, demonstrating that it also exhibits local generation bias. All these results show that various models suffer from such bias.
> * **Theoretical Study.** We further explain the insight behind our theoretical analysis in general response 4 and revised draft. We find that when marginal distributions of tokens are nearly independent, the trained dynamics of score matching is provably biased towards local information in MLP. The model's weight receives a very sparse gradient, only getting significant information within local region. This means the gradient descent will provably lead to a denoising function which primarily focusing on local region despite global receptive field.
> * **New Paradigm.** We agree that a new training paradigm might mitigate hallucination. But that is beyond our scope. We are not claiming that ***any*** paradigm could not solve the hallucination issue (see general response 1). We are focused on why current paradigm without explicit prior knowledge and direct symbolism modeling on text content suffers from this hallucination, and how it is related to the underlying distribution's property and gradient descent.
>
> Given these theoretical results and empirical validation, could you please inform us what are your further concerns? Although the revision period will soon be closed so we cannot conduct new experiments, we are still more than happy to hear constructive advices from you.

---

> ### Comment · Reviewer_yRki · 2024-11-29
>
> Indeed, hallucination is not an already solved issue, but has been greatly alleviated by new architectures. Overall, I still hold the opinion that the empirical evaluation is too weak to support the central claim. Therefore, I would like to keep my rating.

---

> > ### Author Response · Authors · 2024-11-29
> >
> > Thank you for your further response. We are sorry that our expression causes some misunderstanding. The sentence
> > > ... they (hallucination issue) stem from fundamental training dynamics rather than
> > architectural limitations.
> >
> > does not mean that we aim to prove that hallucination is inevitable due to training dynamics. And we never claimed that it is impossible for any future model to solve hallucination issue (maybe we can just leave out the text content as a separate generation process by "language model + prompt rendering" pipeline).
> >
> > **What we want to emphasize is that, the local generation bias is not a result of local receptive field or architectural limitation, but a phenomenon that happens to almost all current models that equipped with global attention capacity.** We have demonstrated in general response 1-3 that even modern diffusion model like FLUX1 and SD3.5 still has high LDR, meaning they generate individual token independently and exhibit such bias.
> >
> > This mechanism is firstly discovered by our paper. We believe this finding still contributes a lot to our current understanding of generative models. In our future version, we will add more constraints to this claim to avoid the misunderstanding that we claim no architectural change could solve the problem. The new statement would be
> > > hallucination issue is mostly correlated to local generation bias, which stems from **current model's** training dynamics rather than architectural limitation in receptive field.
> >
> > Would you think this to be a better claim? (Since revision period has expired, we cannot edit current draft.)

---

> > > ### Comment · Reviewer_yRki · 2024-11-30
> > >
> > > Thank you for your further response. It is indeed clearer now. I would like to raise my rating to 6.

---

> > > > ### Author Response · Authors · 2024-11-30
> > > >
> > > > Thank you very much for your constructive advice and affirmation. We highly appreciate your time and valueable feedback. The modifications would be seen in our final version.

---

### Author Response · Authors · 2024-11-20
**General Response to Common Questions. Part(2/2)**

## 3. More Experiments and LDR Analysis in Wild.

More than one reviewer ask about experiment details, especially figure 5. Here we update the figure and explain several confusions. The total step for UNet is $T=100$ and DiT is $T=250$. The total number of parameters of attention augmented UNet of channels (64,128,256,512) (35.7 M) and DiT-S (33 M) are comparable. It should have been in appendix and we apologize for our negligence.

**Model architecture does play a role, since DiT can eventually overfit well but UNet can not. But both models undergo a hallucination phase and shares similar local generation bias for extensive training steps. We aim to understand the mechanism.**

In the appendix, there are experiments with more data settings (English words, Chinese characters' images) and illustrative visualizations. We recommend interested reviewers to read for more information.

As requested by the reviewers, we also conduct proposed LDR analysis on current models such as SD3.5 and FLUX1. However, Exact calculation the Jacobian matrix $J_{R,\theta}(x)$ requires large memory consumption for these enormous models with billions of parameters. Till the date we post this response, we have not found GPU with such memory capacity (>100 GB). Instead, we use zeroth order approximation to test LDR. Note that
$E_{\epsilon}[\|f(x_0+\epsilon)-f(x_0)\|^2]\approx E_{\epsilon}\langle \epsilon, \nabla_{x_0}f(x) \rangle^2$. Therefore, we can set $\epsilon_1 \sim \mathcal{N}(0,\epsilon^2 I_d)$ and get $E_{\epsilon_1}\langle \epsilon_1, \nabla_{x_0}f(x) \rangle^2=\epsilon^2 \| \nabla_{x_0}f(x)\|^2$. Then set $\epsilon_2\sim \mathcal{N}(0,\epsilon^2 P_RP_R^{\top})$ to get $E_{\epsilon_2}\langle \epsilon_2, \nabla_{x_0}f(x) \rangle^2=\epsilon^2 \| P_R\nabla_{x_0}f(x)\|^2$. Taking ratio of these two terms, we obtain an approximation of LDR. We test the average LDR with prompting "A blackboard with formulas". Total reverse timesteps are 50 and we independently sample 20 $\epsilon$ for total 10 seeds to estimate the expectation.  The region $R$ is selected as the central token which contains text. Here are the approximated LDR results:

| Timestep | 50       | 40       | 30       | 20       | 10       |
|----------|----------|----------|----------|----------|----------|
| FLUX1    | 0.9253   | 0.8078   | 0.7992   | 0.8208   | 0.7994   |
| SD3.5    | 0.9692   | 0.8283   | 0.7912   | 0.5726   | 0.4534   |

Apart from SD3.5 has a small LDR near the end (we do not know why), most LDR is relatively high, which corroborates our finding. These results will be added in future manuscript versions with exact LDR calculation.

## 4. Theory Explanation and How it Supports Central Claim.

The theoretical analysis aims to give a comprehensive explanation under a simple yet non-trivial setting. We admit that it may not directly apply to real-world complicated models, i.e. UNet, DiT, but the point is to provide insight for why training dynamics prefers a local-generation manner. While it is hard to prove things for general models due to technical difficulties, the two-layer network model is widely used in previous theoretical literature for understanding learning phenomenon (e.g. arxiv 2207.08799, 2206.15144, 2110.13905 etc.) **We find that when marginal distributions of tokens are nearly independent, the trained model of score matching is provably biased towards local information.** The model's weight receives a very sparse gradient, only getting significant information within local region. This means the gradient descent will provably lead to a denoising function which primarily focusing on local region despite global receptive field. This insight is also verified via LDR analysis for real-world models (like UNet and DiT), which indeed denoise different tokens' region independently.

---

### Author Response · Authors · 2024-11-20
**General Response to Common Questions. Part(1/2)**

We thank all the reviewers for their time and effort. Since multiple reviewers raise similar concerns, we post the common response in this comment section and recommend all reviewers to read it.

## 1. Understanding Mechanism, Not Impossibility Result.

**The main point of our work is *not* to disprove the possibility of diffusion models to generate any valid typography content, but to understand the universal mechanism behind nonsensical texts.**
We admit that with sufficient (but tremendous) amount of data, training steps and parameters in synthetic setting, model can eventually capture the global relation by overfitting(namely memorizing all the training data). Adding prompt to specify text content then rendering can also circumvent this problem because the network does not need to care about content. But this is orthorgonal to our main focus, which aims to understand the mechanism before such overfitting phase happens. In real practice, hallucinated texts are actually still ubiquitous in modern diffusion models (see 2), since the training stops far before overfitting even if we have trained for a very long time. Moreover, our finding, the identification of local generation bias has its meaning beyond text hallucination.

## 2. Does Modern Generative Models Really Solve Text Hallucination?

We conduct experiment on recent SOTA generative models that claimed to solve text issue, including Stable Diffusion 3.5 and FLUX1, to test the performance of unconditional text generation with following prompts:
* "A blackboard with formulas."
* "A newspaper reporting news."
*  "A piece of calligraphy art."


We also try prompts specifying the texts, i.e. "A paper saying 'To be or not to be, it is a question'. " StableDiffusion 3.5 and FLUX1 can generate text that is overally correct, but still has spelling problem or missing words. All these results are depicted in our Appendix page 18~20. **It can be seen that the generated text is still not readable, demonstrating that text hallucination in unconditional generation still is yet far from resolved.**

---

### Author Response · Authors · 2024-11-26

Dear Reviewers,

We have revised the paper and added plentiful additional results to address your questions in comments. Since the rebuttal period is closing very soon, can you please check the response to see whether it mitigates your concerns? We would greatly appreciate that and are happy to further answer your questions!

---

### Meta-Review · Area_Chair_sVUB · 2024-12-21

**Metareview:**

This paper have received ratings of 6, 6, 6, 6, 6, where the reviewers generally agree to accept this paper. In this paper, the authors identifies local generation bias as a key cause of text hallucination in diffusion models, where symbols are locally coherent but globally nonsensical. Through theoretical analysis and empirical studies, the authors demonstrate that this bias stems from training dynamics, leading models to overly rely on local information and neglect global dependencies/consistency. The proposed Local Dependency Ratio (LDR) quantifies this bias, offering insights to address hallucination across modalities and improve generative models.

Strengths:
- Identify the local generation bias: the paper introduces the concept of local generation bias, a factor contributing to text hallucination in diffusion models. This is a good contribution to understanding generative model artifacts.
- Empirical and theoretical rigorness: The study combines experimental insights with a robust theoretical framework. The introduction of the Local Dependency Ratio (LDR) as a metric is innovative and well-executed.
- Broad applicability: The analysis applies to diverse architectures (e.g., UNet, DiT) and various datasets, enhancing the generality of the findings. The results extend beyond textual data, providing insights into hallucination across modalities.

Areas for Improvement:
- Clarity, especially in theoretical presentation: some sections on training dynamics and saddle point behavior could be further simplified or better illustrated for accessibility.
- Real-world impact not clear. While real-world applicability was discussed, more concrete examples or applications could further validate the findings.
- Exploring mitigation strategies for local generation bias: while the paper identifies/analyzes the issue, exploring more effecitive methods to further reduce this bias during training would enhance its practical utility and impact.

This paper stands out for its scientific rigorness and the novelty of its contributions. The authors addressed all significant reviewer concerns, strengthening the case for the paper's impact. This paper is also very timely, in terms of understanding generative model artifacts---a critical area in machine learning research---and provides tools and insights that can influence future work on model design and evaluation. These collectively merit acceptance.

**Additional Comments On Reviewer Discussion:**

During discussion, the authors highlighted several strengths of the paper, including its identification of local generation bias, the rigorous combination of theoretical and empirical analyses, and the broad applicability of its findings across modalities. The reviewers reached a consensus that the manuscript makes a good contribution to understanding hallucinations in diffusion models.

One recurring concern, however, is the clarity of certain theoretical explanations, particularly regarding the training dynamics and saddle point behavior. While this was initially flagged as a limitation, the authors' rebuttal addressed these points by providing detailed clarifications.

Another point of discussion involved the generalizability of the findings to real-world scenarios. Reviewers appreciated the inclusion of experiments on real-world datasets during the rebuttal phase, which strengthened confidence in the applicability of the results.

---

### Decision · Program_Chairs · 2025-01-22

Accept (Poster)